# Ubiquitylation-dependent Rap2 activation regulates lamellipodia dynamics during cell migration

Andrew Neumann[1], Revathi Sampath[1,2], Emily Mayerhofer[1], Valeryia Mikalayeva[2], Vytenis Arvydas Skeberdis[2], Ieva Sarapinienė[2] and Rytis Prekeris[1,*]

## ABSTRACT

Cell migration is a complex process hallmarked by front-to-back cell polarity that is established by the highly dynamic actin cytoskeleton. Branched actin polymerization creates a lamellipodium at the leading edge of the cell, while the contractile acto-myosin cytoskeleton is present at the lagging edge. Rap2, a Ras GTPase family member, has previously been reported to localize to the lamellipodium as a result of ubiquitylation by a Rab40–Cullin5 E3 ubiquitin ligase complex (Rab40/CRL5). However, how Rap2 functions and how ubiquitylation targets Rap2 to the lamellipodium remained unclear. Here, we demonstrate that Rap2 is recruited to retracting lamellipodia ruffles where it inhibits RhoA, likely through interactions with ARHGAP29, and regulates lamellipodia dynamics, thus facilitating cell migration. Furthermore, using a variety of genetic and pharmacological techniques, we show that Rab40/CRL5-dependent ubiquitylation is required for guanine-nucleotide-exchange factor (GEF)-dependent Rap2 activation, a necessary step for Rap2 targeting to the lamellipodium membrane. As such, we demonstrate how this unique ubiquitylation and activation of Rap2 regulates lamellipodia actin dynamics during cell migration.

KEY WORDS: Cell migration, Rap2, Ubiquitylation

## INTRODUCTION

Cell migration is a complex process essential for multiple functions including organism development and immune responses (Franz et al., 2002; Bravo-Cordero et al., 2012; Trepat et al., 2012). A plethora of factors must be tightly coordinated for proper cell migration. One of which, the actin cytoskeleton, is essential for providing the dynamic structures necessary for cellular locomotion (Neumann and Prekeris, 2023; Schaks et al., 2019). The actin cytoskeleton polymerizes to form multiple uniquely constructed structures that, in a migrating cell, are spatially segregated to create front-to-back polarity. Simply put, the front, or leading edge, of the cell is characterized by Arp2/3-dependent branched actin filaments, which push the front of the cell forward, creating a ruffling structure known as the lamellipodium. The rear, or lagging edge, is canonically defined by focal adhesion-attached linear actin filaments connected by non-muscle myosin II (NMII) motors to form acto-myosin stress fibers. When activated, NMII facilitates acto-myosin stress fiber contraction, which pulls the lagging edge forward (Suraneni et al., 2012; Mullins et al., 1998; Fregoso et al., 2022; Bisi et al., 2013; Naumanen et al., 2008; Kolega, 2006; Hotulainen and Lappalainen, 2006). These distinct actin structures must be tightly spatiotemporally regulated to maintain cell movement in the appropriate direction.

Regulation of the actin cytoskeleton is largely facilitated by small monomeric GTPases. GTPases act as molecular switches that cycle between an active, GTP-bound state and an inactive, GDP-bound state. Activation of GTPases is facilitated by guanine-nucleotide-exchange factors (GEFs) and inactivation by GTPase-activating proteins (GAPs) (Bos et al., 2007). The Rho family of GTPases is largely accepted as the master regulators of actin during cell migration. Among several Rho GTPases, Rac1 is generally considered to be the major regulator of leading edge actin dynamics. Rac1 activates the Arp2/3 complex, resulting in branched actin polymerization and lamellipodium formation (Kurokawa et al., 2004; Kraynov et al., 2000; Suraneni et al., 2012; Simanov et al., 2021; Molinie and Gautreau, 2018; Gorelik and Gautreau, 2015). Oppositely, RhoA is accepted as the main regulator of lagging edge actin structure by activating its effectors, which stimulate NMII activity, resulting in acto-myosin stress fiber contraction and focal adhesion activation (Wong et al., 2023; Watanabe et al., 1999; Kurokawa and Matsuda, 2005; Julian and Olson, 2014; Naumanen et al., 2008; Kolega, 2006; Hotulainen and Lappalainen, 2006; Chrzanowska-Wodnicka and Burridge, 1996; Ridley and Hall, 1992). Precise coordination of these major events of leading edge extension and lagging edge retraction provide the basic mechanics for cell locomotion. Thus, regulation of RhoA and Rac1 localization and activation through GEFs and GAPs has become a major focus of study in the field of cell migration.

Despite the importance of Rho GTPases, other GTPases have also been shown to regulate actin dynamics associated with cell migration. Specifically, the Ras family of GTPases is known for regulating actin cytoskeleton dynamics and cell migration, largely through the Ras subfamily of GTPases (HRAS, KRAS, NRAS), which have been found to be mutated in individuals with cancer and drive cancer metastasis (Fuentes-Calvo et al., 2013; Collins et al., 2023; Bos et al., 2007; Prior et al., 2012). Less commonly studied among the Ras GTPases is the Rap GTPase subfamily (comprising Rap1a, Rap1b, Rap2a, Rap2b, Rap2c). Rap1 and Rap2 have been shown to regulate formation of endothelial barrier monolayers; however, the function of these proteins remains less understood in other cellular contexts such as cell migration (Post et al., 2015; Pannekoek et al., 2013). Rap1 and Rap2 have both been previously implicated in regulating cell migration by mediating focal adhesion formation through integrin trafficking (Rothenberg et al., 2023; Reedquist et al., 2000; Stanley et al., 2012; Miertzschke et al., 2007; Dong et al., 2012). Nevertheless, although we have shown the necessity of Rap2 in random cell motility (Duncan et al., 2022) and

[1]Department of Cell and Developmental Biology, School of Medicine, University of Colorado Anschutz Medical Campus, Aurora, CO 80045, USA. [2]Institute of Cardiology, Lithuanian University of Health Sciences, Kaunas LT-50162, Lithuania.

*Author for correspondence (rytis.prekeris@ucdenver.edu)

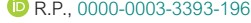 R.P., 0000-0003-3393-1963

others have previously shown the importance of Rap2 in migration upon chemo-stimulation (McLeod et al., 2002; Pöling et al., 2011; Gera et al., 2017), the molecular basis of Rap2 function and regulation during cell migration remains largely unclear.

Our previous work has shown that Rap2 is a substrate of a Rab40–Cullin5 E3 ubiquitin ligase complex (Rab40/CRL5). Our data suggest that this complex can mono-ubiquitylate Rap2 at K117, K148 and K150 (Duncan et al., 2022), although further research will be needed to fully determine the type of ubiquitin-based modifications on Rap2. What is clear, however, is that these ubiquitin modifications are necessary for Rap2 to be targeted to the lamellipodium plasma membrane. Furthermore, ubiquitylation facilitates Rap2 to be in its active, GTP-bound state. Inhibition of Rab40/CRL5-dependent ubiquitylation of Rap2 results in inactivation of Rap2 molecules, which consequently accumulate in and are degraded by lysosomes (Duncan et al., 2022). Results from this study have shown that Rap2 is dynamically regulated in a unique way, which ascribed the need for further study into Rap2 regulation and function. Specifically, we sought to understand both functional and regulation-related aspects of Rap2 by answering the questions of how ubiquitylation of Rap2 by Rab40/CRL5 facilitates its conversion to a GTP-bound (active) state, how ubiquitylation leads to Rap2 targeting to the lamellipodium and plasma membrane, and how ubiquitylated Rap2 affects cell migration.

Here we provide answers to these questions by investigating a functional role of Rap2 in actin dynamics at the lamellipodium and discovering the sequence of events by which ubiquitylation promotes Rap2 localization to the membrane. We demonstrate that Rap2 facilitates cell migration by inhibiting RhoA at the lamellipodium, thus contributing to the establishment of front-to-back polarity in migrating cells. Importantly, we demonstrate that Rap2 is recruited to the lamellipodium plasma membrane during ruffle retraction, where it appears to inhibit RhoA through recruitment and/or activation of ARHGAP29. Thus, Rap2 recruitment to the retracting lamellipodium limits the extent of ruffle retraction and regulates lamellipodia dynamics and cell migration. We also show that Rab40/CRL5 ubiquitylation of Rap2 is essential for its function in regulating RhoA activity and its downstream functions during migration. Furthermore, using a combination of techniques to inhibit membrane trafficking and alter Rap2 activity, we demonstrate that Rab40/CRL5 ubiquitylation of Rap2 at K117, K148 and/or K150 is the first step at the membrane necessary for Rap2 activation and maintenance at the lamellipodium plasma membrane. Accordingly, we propose a model in which the Rap2 GTPases are regulated in their migratory functions through ubiquitylation by the Rab40/CRL5 complex. Ubiquitylated Rap2 is activated by Rap GEFs, facilitating its retention at the lamellipodium plasma membrane where it regulates leading edge dynamics by inhibiting RhoA-driven retraction of actin ruffles.

## RESULTS
### Rap2 is required for cell migration and lamellipodia formation
Although Rap2 has been previously shown to be essential for MDA-MB-231 cell migration and invasion (Duncan et al., 2022), how and when Rap2 regulates membrane and actin dynamics at the lamellipodium in migrating cells remains to be fully understood. To determine the molecular machinery governing the role of Rap2 during migration, we created a Rap2a, Rap2b and Rap2c co-knockout (Rap2-KO) MDA-MB-231 line using CRISPR/Cas9 (Duncan et al., 2022). In order to better understand what aspect of cell migration is mediated by Rap2, we performed individual cell random migration assays using time-lapse imaging. Consistent with the involvement of Rap2 in regulating cell migration, Rap2-KO

cells moved slower and moved a shorter distance as compared to control cells (Fig. 1A–C; Movies 1 and 2).

Although these defects in migration have been previously observed, we sought to further uncover what molecular mechanisms drive the migration defects in Rap2-KO cells. Accordingly, we used immunofluorescence microscopy (IF) to visualize the actin cytoskeleton in Rap2-KO cells. Control MDA-MB-231 cells exhibited clear front-to-back polarity hallmarked by a wide leading lamellipodium with actin ruffles and a discrete and narrow lagging edge (Fig. 1D). In contrast, Rap2-KO cells did not have clear leading or lagging edges and exhibited a visible increase in stress fibers and ventral surface area (Fig. 1D,E). The loss of clear leading and lagging edges suggested a loss of cell polarity. Cell aspect ratio can be used as a measurement to analyze front-to-back polarization (Wong et al., 2023). Consistent with the role of Rap2 in the establishment of front-to-back polarity, Rap2 knockout led to a decrease in aspect ratio (Fig. 1F).

We next sought to understand how a morphological loss of polarity affected other migratory structures. As dynamic, branched-actin-enriched lamellipodia are hallmarks of polarized mesenchymal migration (Suraneni et al., 2012; Schaks et al., 2019), we used IF to visualize cortactin, an actin-binding protein enriched at areas of Arp2/3-dependent branched actin filaments, as a molecular marker of dynamic lamellipodia (Weed et al., 2000; Kaksonen et al., 2000). As shown in Fig. 1G,H, only ~50% of Rap2-KO cells had lamellipodia-like structures with visible cortactin enrichment, indicating that most Rap2-KO cells are incapable of creating or sustaining a lamellipodium. Of the population of Rap2-KO cells with visible cortactin-enriched domains, the lamellipodia-like structures were significantly smaller and displayed less cortactin enrichment as compared to the controls (Fig. 1I,J). Thus, although some Rap2-KO cells can form lamellipodia-like structures, these lamellipodia are smaller and likely contain less dynamic branched actin, further explaining defects in front-to-rear polarity. Accordingly, our data demonstrate that loss of Rap2 impacts the formation of crucial migratory structures such as lamellipodia, which affects front-to-back polarity in migrating cells.

Although significant, the incomplete loss of cortactin enrichment in Rap2-KO cells suggested that their lamellipodia-like structures might still ruffle and function as a lamellipodium. Accordingly, we next assessed whether changes in lamellipodia dynamics contributed to the observed migratory defects. Stroboscopic analysis of cell dynamics (SACED) was performed using previously established calculations from kymographs generated from time-lapse microscopy of migrating MDA-MB-231 cells, allowing for the analysis of lamellipodium ruffling and extension–retraction dynamics (Hinz et al., 1999; Fig. 2A; Fig. S1A, Movies 3 and 4). Noticeably, whereas Rap2 knockout had little effect on ruffle extension and retraction rates (Fig. S1B–D) it significantly decreased retracting ruffle periodicity as compared to that of control cells (Fig. 2B). Rap2-KO cells also exhibited increased membrane retraction distance, causing lamellipodia to retract more than they extend (Fig. 2C), likely contributing to the inability of Rap2-KO cells to move. In accordance with increased retraction, in all our time-lapse videos, Rap2-KO cells were consistently seen to have large retraction events occur at the lamellipodium during ruffling (Movie 4). In contrast, control cells exhibited constantly ruffling lamellipodia with membranes that extend a longer distance than they retract, thus contributing to forward cell movement (Fig. 2C).

To understand how Rap2 regulates lamellipodia extension–retraction dynamics, we generated an MDA-MB-231 cell line stably expressing GFP-tagged wild-type Rap2a (GFP–Rap2a-WT). Notably, GFP–Rap2a-WT was found to be enriched at the lamellipodium of the cell (Fig. S1E), suggesting the need to understand Rap2 dynamics and

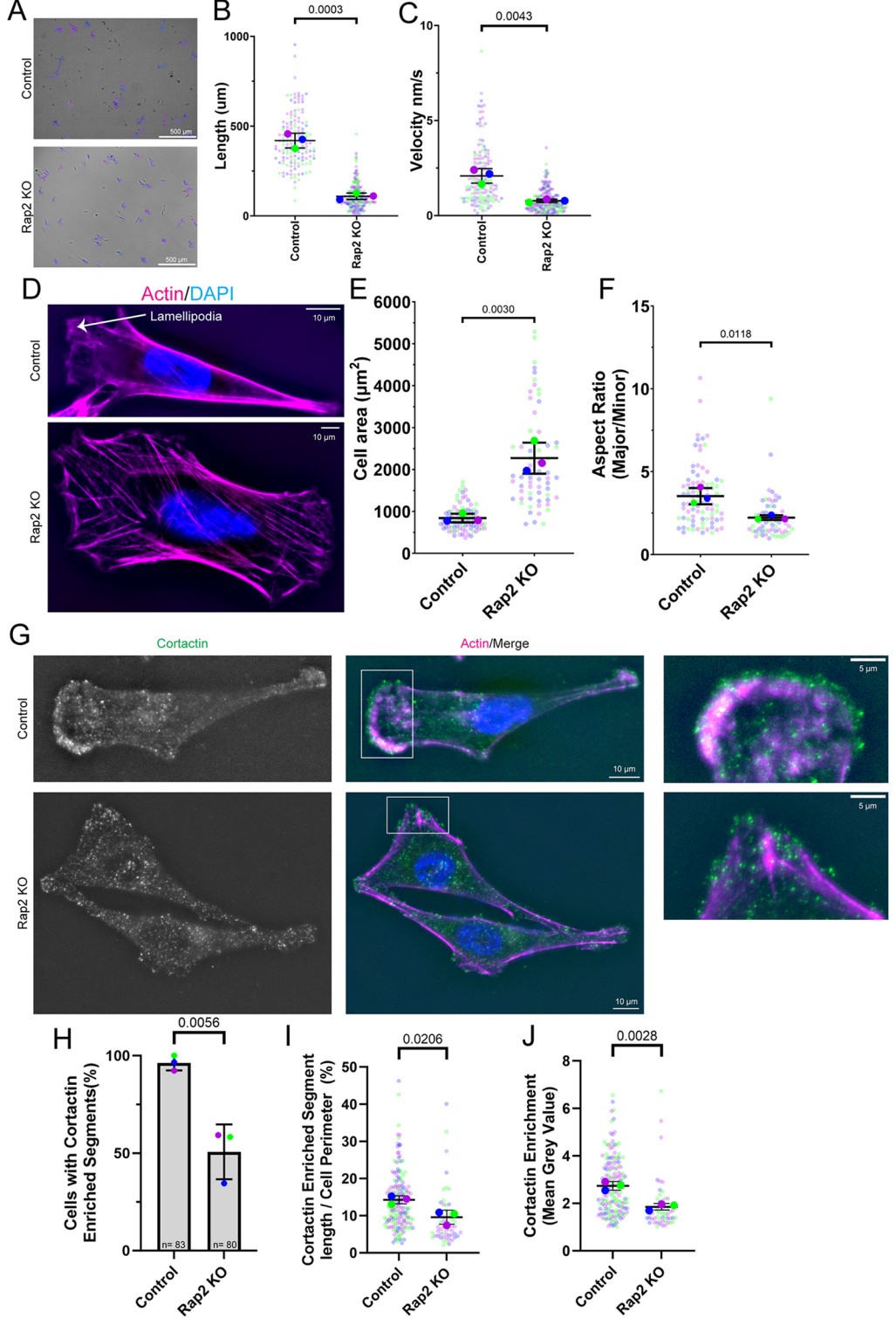

**Fig. 1. Rap2 is required for cell migration and polarized dynamic lamellipodia.** (A–C) Tracking of control and Rap2-KO cells from time-lapse imaging of individual cell random migration assays (Movies 1 and 2). (A) Representative images with cell tracks marked. (B,C) Quantifications (*n*=3 biological replicates; control, 147 total cells; Rap2-KO, 209 total cells; average is mean of biological replicates, error bars represent biological replicate s.d.; unpaired one-tailed Student's *t*-test) show the total length of the track the cells migrated (B) and velocity at which the cells migrated (C). (D) Phalloidin staining of Rap2-KO and control MDA-MB-231 cells to visualize the actin cytoskeleton. (E,F) Quantification (*n*=3 biological replicates; control, 85 total cells; Rap2-KO, 62 total cells; average is mean of biological replicates, error bars represent biological replicate s.d.; unpaired one-tailed Student's *t*-test) of cell area (E) and aspect ratio (F) for cells as in D. (G) Control and Rap2-KO MDA-MB-231 cells were fixed and stained with phalloidin and an anti-cortactin antibody (marker for dynamic lamellipodia). Boxes mark regions shown in enlarged images on the right. (H–J) Quantification (*n*=3 biological replicates; control, 83 total cells, 164 enriched segments; Rap2-KO, 80 total cells, 52 enriched segments; average is mean of biological replicates, error bars represent biological replicate s.d.; unpaired one-tailed Student's *t*-test) of anti-cortactin antibody signal in MDA-MB-231 cells. (H) Percentage of cells with cortactin-enriched segments. (I) Percentage of the individual cell perimeter made up of a cortactin-enriched segments. (J) Enrichment of cortactin in the lamellipodium as compared to the actin cortex.

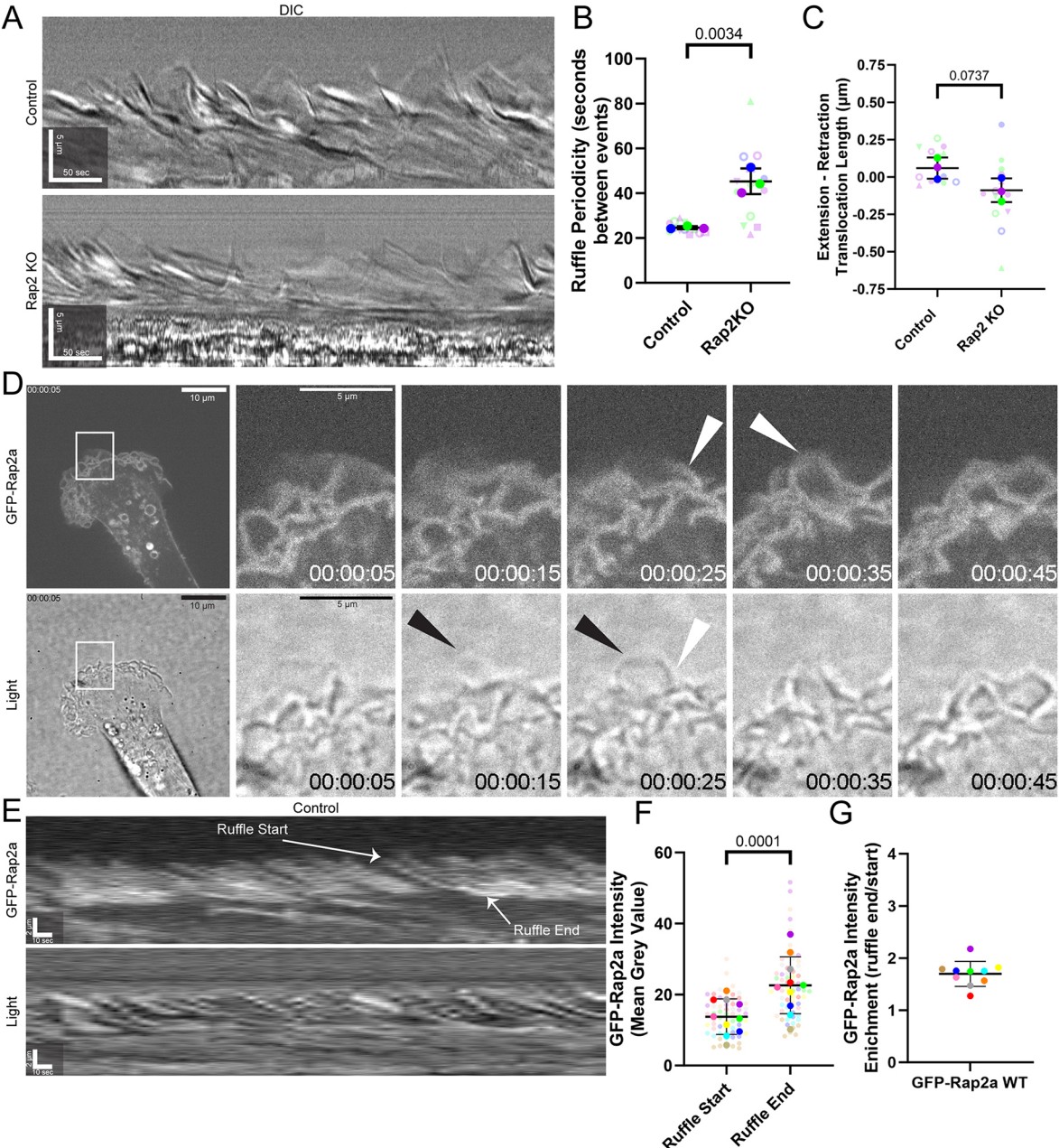

**Fig. 2. Rap2 localizes to and regulates leading edge ruffles.** (A) Kymographs taken from SACED time-lapse imaging of control and Rap2-KO cells (Movies 3 and 4). (B,C) Actin ruffle periodicity (B) and membrane extension and retraction length (C) were measured from SACED kymographs as in A (*n*=3 biological replicates; control, 13 total cells; Rap2-KO, 13 total cells; average is mean of biological replicates, error bars represent biological replicate s.d.; unpaired one-tailed Student's *t*-test. (D) GFP–Rap2-expressing MDA-MB-231 cells were analyzed by time-lapse microscopy (see Movie 5). Shown are selected GFP–Rap2a and transmitted light images taken from time-lapse series (time shown as h:min:s). Boxes indicate regions shown in magnified images on the right. Black arrowheads point to an extending ruffle that lacks GFP–Rap2a. White arrowheads point to a retracting ruffle with accumulating GFP–Rap2a. (E) Kymographs generated from GFP–Rap2a time-lapse series (Movie 5). Ruffle start and ruffle end are labeled as examples of where mean grey value was measured. (F,G) Quantification (*n*=10 cells; 60 total ruffles; average is mean of ruffles from cells, error bars represent s.d. of ruffles averaged per cell; one-tailed paired-Student's *t*-test in F) of mean grey value from the start and end of ruffles from kymographs of GFP–Rap2a dynamics as in E, showing that GFP–Rap2a accumulates during ruffle retraction.

function at the lamellipodium. As such, we analyzed the dynamics of GFP–Rap2-WT recruitment to the lamellipodium using time-lapse microscopy (Movie 5). As shown in Fig. 2D,E, GFP–Rap2a-WT is largely absent at the lamellipodium as the membrane extends forward (Fig. 2D, black arrowheads) and accumulates on the ruffle as it retracts (Fig. 2D, white arrowheads). Interestingly, the fluorescence intensity of GFP–Rap2a-WT increases as the ruffle retracts, indicating an increasing accumulation of Rap2a throughout

the ruffle retraction process (Fig. 2E–G). Increasing Rap2 localization in the retracting lamellipodium suggests that Rap2 might function to terminate lamellipodia ruffle retraction. To test this hypothesis, we also examined the dynamics of lamellipodia ruffles when cells express a constitutively active Rap2 construct (GFP–Rap2a-Q63E; Movie 6). GFP–Rap2a-Q63E-enriched retracting lamellipodia ruffles were observed to be slower, shorter and more disorganized than GFP–Rap2a-WT lamellipodia ruffles. Additionally,

GFP–Rap2a-Q63E ruffle retraction in the lamellipodium was slower and shorter as compared to GFP–Rap2a-WT ruffles (Fig. S2A–D). These results are consistent with the data that Rap2 knockout leads to an increase in lamellipodia retraction distance and a decrease in lamellipodia ruffling periodicity (Fig. 2B,C) and cumulatively suggest that Rap2 functions to terminate lamellipodia ruffle retraction in the lamellipodium.

## Rap2 regulates RhoA in retracting lamellipodia ruffles

Rap2 targeting to the retracting lamellipodia ruffles suggests a distinct role in the regulation of branched actin dynamics. Previously, in neutrophil-like HL-60 cells, Rap2 has been shown to bind β-arrestin 1 at the lamellipodium. Rap2 and β-arrestin 1 separate upon sensing of a chemo-stimulant, allowing β-arrestin 1 to help polymerize branched actin and promote lamellipodium extension (Gera et al., 2017). However, here we observe Rap2 accumulation in retracting ruffles, which is not consistent with its role in β-arrestin 1 binding. We wondered whether Rap2 might influence core actin dynamic regulators, specifically the Rho family of GTPases. RhoA is generally viewed as a positive regulator of actin contractility at the lagging edge of the cell (Hotulainen and Lappalainen, 2006; Ridley and Hall, 1992). However, RhoA has also been shown to be necessary in conjunction with Rac1 activity for lamellipodium ruffling, likely by providing contractile pulses during ruffle retraction (Qian et al., 2024; Heasman and Ridley, 2010; O'Connor et al., 2000; O'Connor and Chen, 2013). As such, we reasoned that the accumulation of Rap2 in retracting ruffles might function to inhibit RhoA. To test this idea, we co-transfected MDA-MB-231 cells with GFP–Rap2a and a biosensor for active RhoA (dTom–2×rGBD; Mahlandt et al., 2021). Consistent with its role in driving ruffle retraction, activated RhoA can be observed to quickly accumulate at the initiation of lamellipodium ruffle retraction, where it colocalizes with GFP–Rap2a (Fig. 3A; Movie 7). Notably, the RhoA biosensor could be observed to localize to retracting ruffles where GFP–Rap2a signal accumulates throughout the retraction process (Fig. 3A, white arrowheads). This colocalization was observed consistently across ruffles throughout the time-lapse series (Fig. 3B). To test whether this colocalization was indicative of Rap2 regulation of RhoA, we expressed the RhoA biosensor in control and Rap2-KO cells. Consistent with the role of RhoA in mediating lamellipodium retraction, in control cells, RhoA biosensor fluorescence was enriched in retracting ruffles. Biosensor-enriched ruffles in control cells retracted inwards toward the cell center, and biosensor fluorescence began dissipating when retraction was terminating (Fig. 3C, white arrowheads; Movie 8). In contrast, while in Rap2-KO cells the RhoA biosensor was also observed to be enriched in retracting ruffles, these ruffles had increased biosensor levels as compared to those of control cells (Fig. 3C,D; Movie 9) – an indication of increased levels of activated RhoA at the lamellipodium. Furthermore, RhoA biosensor-containing retracting ruffles in Rap2-KO cells did not always retract directly towards the cell center but instead could be observed moving horizontally across the lamellipodium (Fig. 3C, black arrowheads; Movie 9). This resulted in ruffles where RhoA biosensor fluorescence lasted longer than that in controls and did not quickly dissipate after contraction (Fig. 3E). All these results indicate that Rap2 might be acting as a RhoA inhibitor in retracting lamellipodia ruffles, thus regulating both the quantity and longevity of RhoA activity at retracting ruffles.

While showing that Rap2 inhibits RhoA activity, we sought to confirm this interaction by understanding how Rap2 could regulate RhoA. We became interested in ARHGAP29, a putative RhoA GAP that is upregulated in many migratory cancer types and has variants that cause cleft palate (Shimizu et al., 2020; Ripperger et al., 2007;

Meng et al., 2018; Kolb et al., 2020; Jiang et al., 2023; Paul et al., 2017; Liu et al., 2017; Leslie et al., 2012). Structural predictions show that, in addition to its RhoGAP domain, ARHGAP29 also has an F-BAR domain and a diacylglycerol (DAG)-binding C1 domain. Since it has been demonstrated that DAG is enriched at the lamellipodium (Ziemba and Falke, 2018; Aguilar-Cuenca and Vicente-Manzanares, 2014), we predicted that ARHGAP29 might be localized to retracting lamellipodia ruffles where it could function to inhibit RhoA (Fig. 4A). Consistent with this hypothesis, whereas the majority of ARHGAP29 was found to be localized in the cytoplasm, a distinct population of ARHGAP29 could also be observed localized to areas of actin ruffling at the lamellipodium but minimally localized to the lagging edge of the cell (Fig. 4B–D). At these lamellipodia regions, even in multi-polar cells with multiple lamellipodia, ARHGAP29 colocalized with GFP–Rap2a (Fig. 4E,F, white arrowheads), suggesting that Rap2 might interact with ARHGAP29 at the lamellipodium. Notably, whereas Rap2 can be localized across the whole membrane, including the lagging edge (Fig. 3A), the minimal localization of ARHGAP29 to the lagging edge (Fig. 4B,D) suggests that any Rap2–ARHGAP29 interaction is likely specific to the lamellipodium. Previous work in *Caenorhabditis elegans* suggests that Rap2 could interact with a truncated form of ARHGAP29 (Myagmar et al., 2005), although interaction between Rap2 and full-length ARHGAP29 remains to be demonstrated. We tested this potential interaction by using GST–GFP-Trap beads to pull down GFP–Rap2a and test for the presence of endogenous ARHGAP29. Since ARHGAP29 contains two membrane-binding domains (F-BAR and C1), we speculated that membranes might be needed for ARHGAP29 activity and binding to Rap2. To that end, we lysed cells using hypotonic shock and used the resulting membrane-containing lysate in our pulldown assays. We found that GFP–Rap2a could pull down ARHGAP29 (Fig. 4G), suggesting that ARHGAP29 may be a canonical Rap2 effector protein. Importantly, little ARHGAP29 was pulled down when beads without GST–GFP-Trap were used, and no membranes derived from endoplasmic reticulum (negative control) were detected in our blank bead pulldown assays, thus suggesting that ARHGAP29 presence in pulldowns was not a result of membrane fragments pelleting with our beads. Finally, interactions between GFP–Rap2a and ARHGAP29 did require the presence of membranes, as lysing cells with detergent prevented GFP–Rap2a from pulling down ARHGAP29 (Fig. 4H). Based on these data, we hypothesize that binding of the ARHGAP29 F-BAR and C1 domains to the lamellipodium membrane is required for the formation of a stable Rap2 and ARHGAP29 complex. It is also intriguing to speculate that coordination between all these domains is what determines the specificity and spatiotemporal control of ARHGAP29 recruitment to the lamellipodium. However, further research will be needed to fully characterize the molecular machinery that regulates the recruitment of ARHGAP29 to the lamellipodium membrane.

## Loss of Rap2 leads to increased formation of focal adhesion sites and acto-myosin stress fibers

Our data demonstrate that Rap2 is involved in regulating cell polarity and migration in part through regulation of RhoA activity during lamellipodia extension and retraction. However, upon Rap2 knockout, we noticed morphological defects throughout the whole cell that are indicative of increased RhoA activity. One of the well-established functions of RhoA is to regulate stress fiber tension and focal adhesion (FA) site formation (Wolfenson et al., 2011; Mishra and Manavathi, 2021; Julian and Olson, 2014). Rap2 has previously been shown to regulate integrin recycling (Stanley et al., 2012; Miertzschke et al.,

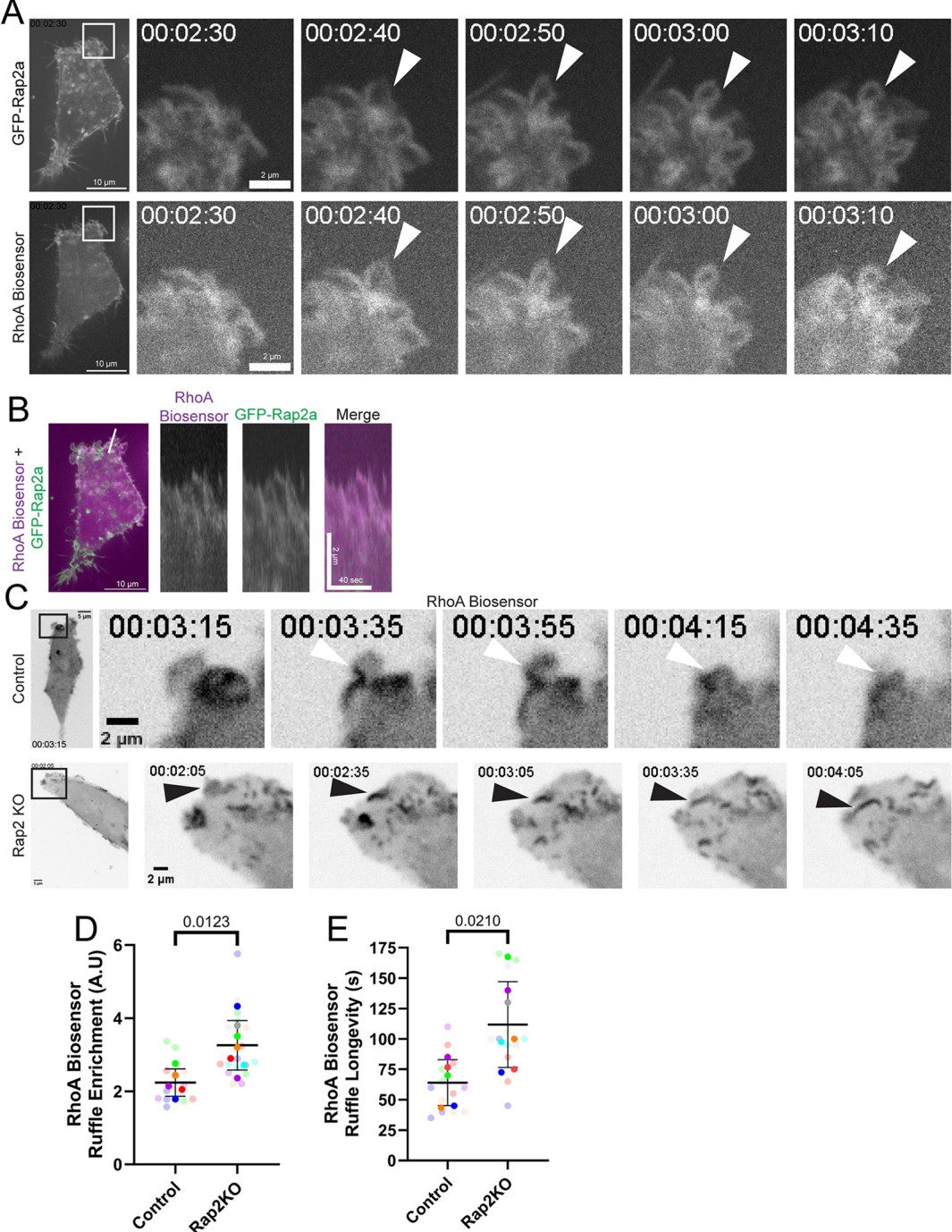

**Fig. 3. Rap2 regulates RhoA activity at the lamellipodium.** (A,B) Time-lapse analysis of MDA-MB-231 cells co-expressing GFP–Rap2a and dTom–2×rGBD (RhoA biosensor) (also see Movie 7). Shown are selected GFP–Rap2a and RhoA biosensor images taken from the time-lapse series of a ruffling lamellipodium (representative of 23 cells imaged). Boxes in A indicate regions shown in magnified images on the right (time shown as h:min:s). White arrowheads show a ruffle where GFP–Rap2a is recruited with the RhoA biosensor. Panel B shows a kymograph displaying the RhoA biosensor and GFP–Rap2a colocalization in ruffles over time. White line in the overview image marks the position of the kymograph. (C–E) Time-lapse analysis of control and Rap2-KO MDA-MB-231 cells expressing dTom–2×rGBD (RhoA biosensor) (Movies 8 and 9). Shown are selected RhoA biosensor images taken from the time-lapse series of a ruffling edge. Boxes indicate regions shown in magnified images on the right (time shown as h:min:s). White arrowheads show RhoA ruffles in control cells retracting vertically; black arrowheads show RhoA ruffles in Rap2-KO cells that retract and move horizontally. Quantification of the enrichment (control, $n=5$ cells, 15 total ruffles; Rap2-KO, $n=7$ cells, 20 total ruffles; average is mean of biological replicates, error bars represent biological replicate s.d.; unpaired one-tailed Student's $t$-test) (D) and longevity (control, $n=5$ cells, 15 total ruffles; Rap2-KO, $n=7$ cells, 13 total ruffles; average is mean of biological replicates, error bars represent biological replicate s.d.; unpaired one-tailed Student's $t$-test) (E) of the RhoA biosensor in retracting ruffles. A.U., arbitrary units.

2007; Dong et al., 2012); however, as we showed Rap2 localization and regulation of RhoA at the lamellipodium, we wanted to see whether effects on FAs in Rap2-KO cells were limited to the leading edge or were present throughout the entire cell. To that end, we transfected control and Rap2-KO cells with GFP–paxillin and analyzed FA dynamics using time-lapse microscopy. As expected,

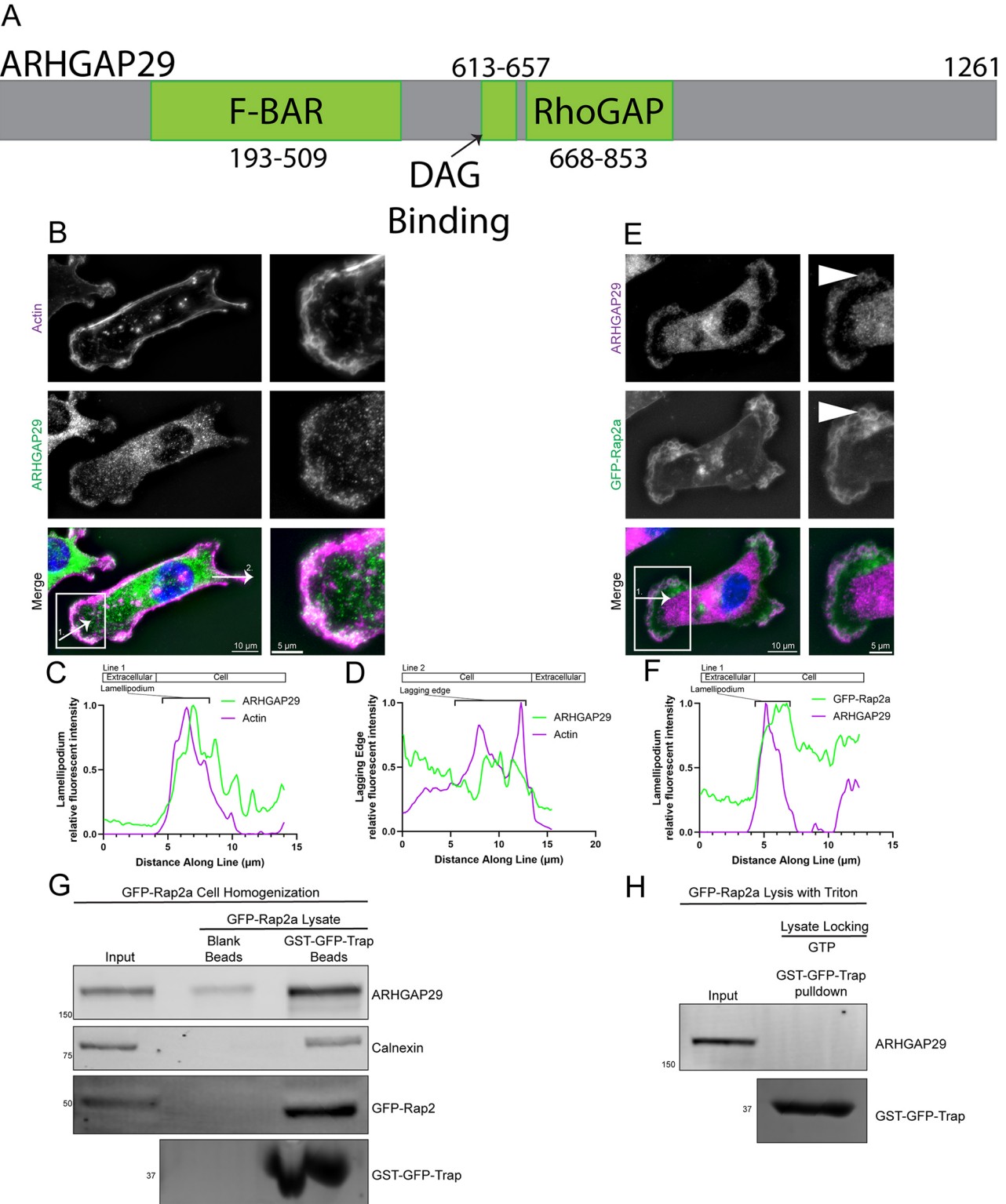

**Fig. 4.** See next page for legend.

migrating control MDA-MB-231 cells formed dynamic nascent FAs, characterized by cytoplasmic FAs with relatively short lifespans and rapid movement across the ventral cell surface during cell migration (Fig. 5A–C; Movie 10). Furthermore, in control cells, nascent FAs at the lamellipodium were characterized by rapid turnover (Fig. 5D, black arrowheads). In contrast, FAs in Rap2-KO cells did not turn over at lamellipodia-like structures, lasted throughout our whole time-lapse

series and moved slowly across the ventral cell surface (Fig. 5A–D; Movie 11). These results are consistent with FAs in Rap2-KO cells having increased stability and having disassembly defects.

To further analyze the FA phenotypes observed in our time-lapse imaging, we performed IF to visualize endogenous FAs using an anti-phospho-paxillin antibody (Bachmann et al., 2022). In Rap2-KO cells, there were more phospho-paxillin-labeled FAs. These FAs were bigger

**Fig. 4. Rap2 interacts with the putative RhoA GAP, ARHGAP29.**
(A) Linear schematic showing the predicted location of the F-BAR, DAG-binding (C1) and RhoGAP domains of ARHGAP29 in green. Grey represents areas predicted to be intrinsically disordered regions. (B–D) Phalloidin labeling of actin and anti-ARHGAP29 staining in a representative control MDA-MB-231 cell. ROI (box and enlarged images, right) shows ARHGAP29 localization to the areas of actin ruffling. Line scans along arrow 1 and arrow 2 drawn in B show ARHGAP29 and actin fluorescence intensity at (C) the lamellipodium and (D) the lagging edge, respectively. (E,F) Anti-ARHGAP29 staining of a representative MDA-MB-231 cell expressing GFP–Rap2a. ROI (box and enlarged images, right) shows the lamellipodium where ARHGAP29 and GFP–Rap2a colocalize. White arrowheads show colocalization of ARHGAP29 and GFP–Rap2a in what appear to be actin ruffles. Line scans along the arrow drawn in E show colocalization of ARGHAP29 and GFP–Rap2a at the lamellipodium (F). Data in B–F representative of 30 cells. (G) GST–GFP-Trap pulldowns of GFP–Rap2a-overexpressing cells burst with 20 strokes of a Dounce homogenizer. ARHGAP29 binding of GFP–Rap2a was assessed by western blotting, and GST–GFP-Trap input was confirmed by Coomassie Blue staining. Calnexin was used as a control to confirm that homogenized membrane segments did not pellet with the beads. Data shown are representative of three experiments. (H) GST–GFP-Trap pulldowns of GFP–Rap2a-overexpressing cells lysed with Triton X-100-containing lysis buffer. Cell lysate was locked with GTP prior to GST–GFP-Trap addition. ARHGAP29 binding of GFP–Rap2a was assessed by western blotting, and GST–GFP-Trap input was confirmed by Coomassie Blue staining. Data shown are representative of three experiments. In G and H, positions of molecular mass markers are indicated in kDa.

than those in control cells (Fig. 5E–G). Additionally, in control cells phospho-paxillin-labeled FAs localized in the cell periphery, in the areas of observed actin ruffling, while also being present intracellularly, anchoring acto-myosin stress fibers. However, in Rap2-KO cells, phospho-paxillin-labeled FAs predominately localized at the cell periphery, at both areas of potential actin extension and areas where they are anchoring stress fibers at the cell periphery (Fig. 5E,H). These data align with our live-cell imaging of GFP–paxillin and further suggest that in Rap2-KO cells, FAs are more stable, do not turn over and have a decreased ability to travel across the cell.

FAs are involved in mediating cytoskeletal tension and allow for the production of contractile forces generated by NMII motors that assemble on FA-anchored stress fibers (Richter et al., 2021; Kolega, 2006). Accordingly, we predicted that the increase in FA number and size in Rap2-KO cells correlated with an increase in contractility of acto-myosin stress fibers. To test this hypothesis, we performed IF microscopy using an anti-NMIIb (MYH10) antibody and analyzed NMIIb distribution along actin stress fibers. As shown in Fig. 5I,J, Rap2-KO cells had greater enrichment of NMIIb on acto-myosin stress fibers as compared to control cells. Thus, our observations regarding FAs in Rap2-KO cells are linked with changes in acto-myosin stress fiber contractility. As downstream effectors of RhoA, increased FA stability and acto-myosin contractility concur with our data suggesting that Rap2 negatively regulates RhoA. Interestingly, we did not observe ARHGAP29 at FAs or stress fibers (Fig. 4B,C), suggesting that Rap2 might regulate RhoA at FAs via a different mechanism. Thus, further work will be necessary to understand how Rap2 regulates FA stability and stress fiber contractility.

### Rap2 ubiquitylation is required for the establishment of front-to-back polarity and regulation of FAs and stress fibers
While our data demonstrate that Rap2 can interact with ARHGAP29 and regulate lamellipodia dynamics, front-to-back polarity, and FA dynamics, we still do not understand the molecular machinery regulating Rap2 activation and targeting to the lamellipodium plasma membrane. For example, while our previous work has suggested that ubiquitylation is needed for Rap2 localization

(Duncan et al., 2022), it remains to be determined whether Rab40/CRL5-dependent ubiquitylation of Rap2 is necessary for its role in regulating these migratory functions. To test whether Rap2 ubiquitylation is required for these cell migration-related functions, we generated MDA-MB-231 Rap2-KO lines rescued by overexpression of ether GFP–Rap2a-WT or a GFP–Rap2a-K3R mutant (which has K117R, K148R and K150R mutations that prevent ubiquitylation by the Rab40/CRL5 complex; Duncan et al., 2022). Microscopy was then used to visualize the actin cytoskeleton and analyze the formation of front-to-back polarity. The loss of front-to-back polarity observed in Rap2-KO lines was mostly restored to control cell levels by expressing GFP–Rap2a-WT, whereas GFP–Rap2a-K3R expression failed to rescue any of these defects (Fig. 6A–C and Fig. 1D–F).

Next, we sought to understand whether Rap2 ubiquitylation is also needed for the function of Rap2 in regulating FAs and actin stress fibers. To that end, we used an anti-phospho-paxillin antibody to analyze the size and number of FAs in Rap2-KO cells expressing GFP–Rap2a-WT or GFP–Rap2a-K3R. Whereas GFP–Rap2a-WT overexpression rescued the Rap2 KO-induced FA defects, overexpression of the GFP–Rap2a-K3R mutant failed to rescue any FA phenotypes (Fig. 6D–F and Fig. 5E–G).

Because GFP–Rap2a-K3R expression was unable to rescue morphological or FA defects, we predicted that GFP–Rap2a-K3R expression would also be unable to rescue Rap2 knockout-induced loss of migration. Accordingly, we repeated random migration assays with control and GFP–Rap2a-K3R rescue lines. GFP–Rap2a-K3R expression was not seen to rescue cell migration velocity and length (Fig. 6G,H) and mimicked the migration characteristics of Rap2-KO cells (Fig. 1B,C). Notably, we have previously shown that Rap2 constructs that cannot be ubiquitylated by the Rab40/CRL5 complex are targeted to lysosomes for degradation (Duncan et al., 2022). This trend is also observed in our Rap2-KO rescues lines, where GFP–Rap2a-K3R does not express as well as GFP–Rap2a-WT (Fig. 6I). Although this variation in expression could impact the rescue of Rap2-KO phenotypes, both GFP–Rap2a-WT and GFP–Rap2a-K3R rescue lines were observed to express GFP–Rap2a at a greater level than the endogenous Rap2 expressed in a control line (Fig. 6I). As such, expression levels of both rescue GFP–Rap2a constructs should have minimal effect on our rescue phenotypes. Thus, our data show that ubiquitylation is necessary for Rap2 function during cell migration. Accordingly, we needed a better understanding of how Rab40/CRL5 ubiquitylation at K117, K148 and/or K150 regulates Rap2 function.

### Ubiquitylated Rap2 is enriched at the plasma membrane and lamellipodia
Because of the importance of Rap2 ubiquitylation in regulating cell migration, we next sought to visualize the subcellular localization of ubiquitylated Rap2. Our previous work has shown that inhibition of Rap2 ubiquitylation causes Rap2 localization to shift from the plasma membrane and lamellipodia to lysosomes (Duncan et al., 2022), leading to predictions that ubiquitylated Rap2 is likely present at the lamellipodium and plasma membrane. Consistent with this hypothesis, we have also shown that Rab40/CRL5 is present at the lamellipodium of migrating cells (Duncan et al., 2022; Linklater et al., 2021). However, these predictions fail to directly visualize the localization of ubiquitylated Rap2, leaving alternative explanations for the role of Rap2 ubiquitylation in trafficking and localization. To address this knowledge gap, we performed a proximity ligation assay (PLA) using anti-GFP and anti-ubiquitin antibodies in MDA-MB-231 cell lines overexpressing either GFP–Rap2a-WT or GFP–Rap2a-K3R. Consistent with our previous reports that Rab40/CRL5

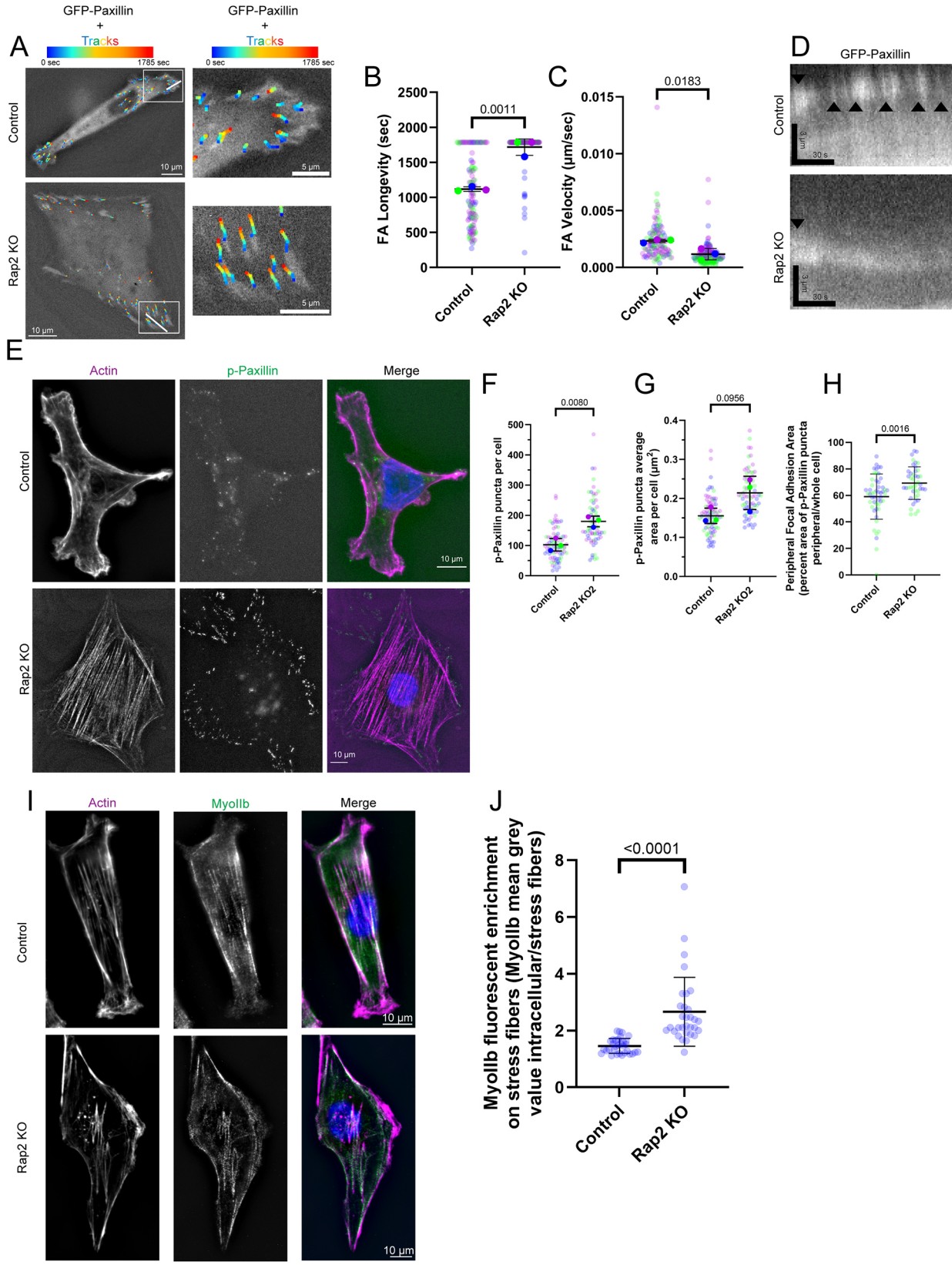

**Fig. 5.** See next page for legend.

ubiquitylates Rap2 on K117, K148 and/or K150, cells expressing GFP–Rap2a-WT had more PLA puncta than cells expressing GFP–Rap2a-K3R (Fig. 7A–C). Importantly, GFP–Rap2a-K3R-expressing

cells had similar levels of PLA puncta to that of cells assayed as controls using only the anti-GFP or anti-ubiquitin antibodies (Fig. 7A–C; Fig. S3A). Furthermore, the greater number of PLA

**Fig. 5. Rap2 regulates the formation and dynamics of focal adhesions.**
(A–D) Time-lapse analysis of MDA-MB-231 cells expressing GFP–paxillin.
(A) Tracks of paxillin puncta movement over time are overlaid on top of the
first image taken. Boxes indicate regions shown in magnified images on the
right. White lines indicate positions of kymographs shown in D.
(B,C) Quantification [*n*=3 cells; control, 115 total puncta; Rap2-KO, 140 total
puncta; average is mean of biological replicates (cells), error bars represent
biological replicate s.d.; unpaired one-tailed Student's *t*-test] of paxillin
puncta longevity (B) and velocity (C). (D) Kymographs generated from
GFP–paxillin time-lapse series (Movies 10 and 11). Black arrowheads mark
the appearance of new FAs as marked by GFP–paxillin signal. (E) Control
and Rap2-KO MDA-MB-231 cells were stained with phalloidin, to label actin,
and with an antibody that detects phospho-paxillin (p-Paxillin).
(F–H) Quantification of FAs from cells as shown in E. Panel F shows the
average number of phospho-paxillin puncta per cell and panel G shows the
average area of phospho-paxillin puncta per cell (*n*=3 biological replicates;
control, 84 cells; Rap2-KO, 71 cells; average is mean of biological replicates,
error bars represent biological replicate s.d.; unpaired one-tailed Student's
*t*-test). Panel H shows the distribution of phospho-paxillin puncta area in the
periphery per cell (control, *n*=51 cells; Rap2-KO, *n*=43 cells; average is
mean of cells, error bars represent s.d.; unpaired one-tailed Student's *t*-test).
(I) Control and Rap2-KO MDA-MB-231 cells were stained with phalloidin and
an anti-MyoIIb antibody. (J) Quantification of cells as in I, showing the
enrichment of MyoIIb fluorescence density on acto-myosin stress fibers
(control, *n*=32 cells; Rap2-KO, *n*=31 cells; average is mean of cells, error
bars represent s.d.; unpaired one-tailed Student's *t*-test).

puncta in GFP–Rap2a-WT-expressing cells was not a result of
change in cell size (Fig. S3B). The few PLA puncta observed in
GFP–Rap2a-K3R-expressing cells indicate that mutation of K117,
K148 and/or K150 of Rap2 reduces the association of Rap2 with
ubiquitin, likely by preventing ubiquitylation of Rap2 by the Rab40/
CRL5 complex.

Next, we asked where PLA puncta localize in cells expressing
GFP–Rap2a-WT to assess the localization of ubiquitylated Rap2.
We found that more PLA puncta were associated with the lamellipodia
membrane in cells expressing GFP–Rap2a-WT than in cells
expressing GFP–Rap2a-K3R (Fig. 7D). Additionally, PLA puncta
in GFP–Rap2a-WT cells were also observed on the ventral cell
membrane (Fig. 7B). These observations suggest that Rap2 is
associated with ubiquitin on the plasma membrane, likely as a result
of Rab40/CRL5 ubiquitylation. Furthermore, we have shown that
Rap2 is active when ubiquitylated (Duncan et al., 2022), and here we
show a role for Rap2 function in regulating RhoA at the lamellipodium
(Fig. 3). In accordance with Rap2 function at the lamellipodium, we
found an enrichment of PLA puncta specifically at the lamellipodium
and areas of membrane ruffling in GFP–Rap2a-WT-expressing
cells (Fig. 7E). Notably, as seen in our Rap2-KO rescue
experiments, GFP–Rap2a-WT expresses better than GFP–Rap2a-
K3R, which is rapidly degraded in lysosomes (Duncan et al., 2022),
though this difference is minimal at early passages (Fig. S3C). As
such, these PLA experiments, and the experiments reported below,
were performed starting with early passage cells. Changes in data
reproducibility were not observed as experiments were performed and
cell passage number increased and GFP–Rap2a-K3R expression
decreased, thus indicating that differences in expression between
GFP–Rap2a-WT and GFP–Rap2a-K3R did not affect these results.
As such, these PLA data suggest that the association of Rap2 with
ubiquitin requires Rab40/CRL5 ubiquitylation of Rap2 at the
membrane and areas of actin ruffling.

## Ubiquitylation facilitates Rap2 retention at the plasma membrane

Our PLA experiment suggests that ubiquitylated Rap2 is localized
to the lamellipodium and plasma membrane. Furthermore, we

have shown that the loss of ubiquitylation results in Rap2
accumulation in lysosomes (Fig. 8A; Duncan et al., 2022),
suggesting that ubiquitylation might regulate either Rap2 delivery or
removal (or both) from the plasma membrane. However, the
mechanism by which ubiquitylation regulates these processes
remains unclear. Accordingly, we next sought to understand how
ubiquitylation mediates Rap2 localization to the lamellipodium and
plasma membrane.

To investigate this mechanism, we hypothesized two possible
explanations for the role of ubiquitylation in Rap2 accumulation at
the lamellipodium membrane. First, Rab40/CRL5 ubiquitylation
might mediate Rap2 delivery to the plasma membrane.
Second, this ubiquitylation might inhibit Rap2 removal from the
lamellipodium plasma membrane, increasing its lamellipodium
membrane dwell-time. Previous work has shown that Rap2 is
predominately membrane associated and is removed from the
plasma membrane by macropinocytic vesicles that form at the
actively ruffling lamellipodium (Duncan et al., 2022). Accordingly,
to test these hypotheses, we used multiple techniques to inhibit
Rap2 macropinocytic internalization from the plasma membrane
and assessed the proportion of GFP–Rap2a-WT and GFP–Rap2a-
K3R localized to the plasma membrane (Fig. 8B). First, we
inhibited endocytosis and macropinocytosis by incubating cells
at 4°C. The 4°C treatment resulted in increased accumulation of
both GFP–Rap2a-WT and GFP–Rap2a-K3R on the plasma
membrane (Fig. 8C,D). Notably, moving cells back to 37°C led to
a rapid removal of GFP–Rap2a-K3R from the plasma membrane
and its accumulation in vesicles that we have previously identified
as either endosomes or lysosomes (Fig. 8C,D) (Duncan et al., 2022).

While effective in inhibiting endocytic and macropinocytic
internalization, 4°C treatment has the potential to alter many other
cellular functions such as enzymatic activity and protein interactions.
To more directly inhibit macropinocytosis, we used CK666, the well-
established inhibitor of the Arp2/3 complex. CK666 treatment
inhibits branched actin polymerization, a necessary step in
ruffling-dependent macropinocytosis (Mylvaganam et al., 2021).
As shown in Fig. 8E,F, CK666 inhibition of macropinocytosis
increased plasma membrane accumulation of both GFP–Rap2a-
WT and GFP–Rap2a-K3R. Since inhibition of GFP–Rap2a-K3R
internalization rescues the Rap2a-K3R localization defect,
Rab40/CRL5 ubiquitylation of Rap2 likely does not play a role in
Rap2 trafficking to the plasma membrane. Instead, ubiquitylation
likely functions to inhibit Rap2 internalization, increasing
Rap2 dwell-time at the plasma membrane. Interestingly, although
both 4°C and CK666 treatment increased GFP–Rap2a membrane
accumulation, it was observed that inhibition of GFP–Rap2a
macropinocytosis resulted in a ubiquitous distribution of GFP–
Rap2a across the entire membrane instead of enrichment at the
lamellipodium membrane. We have previously reported that
recycling of macropinocytosed Rap2 endosomes is necessary for
Rap2 enrichment at the lamellipodium (Duncan et al., 2022). Thus,
although ubiquitylation inhibits the internalization of Rap2 from
the membrane, internalization and the subsequent recycling is likely
still necessary for Rap2 localization and function as an inhibitor of
RhoA at the lamellipodium.

It is well established that Rap2 GEFs are located at the plasma
membrane (Bos et al., 2007; Kumar et al., 2018; Sartre et al., 2023).
Thus, it was proposed that targeting Rap2 to the plasma membrane
leads to its binding to Rap2 GEFs, resulting in Rap2 activation. We
have previously shown that the Rap2-K3R mutation blocks its
activation (Duncan et al., 2022), raising an intriguing possibility that
increasing Rap2a-K3R dwell-time at the plasma membrane might

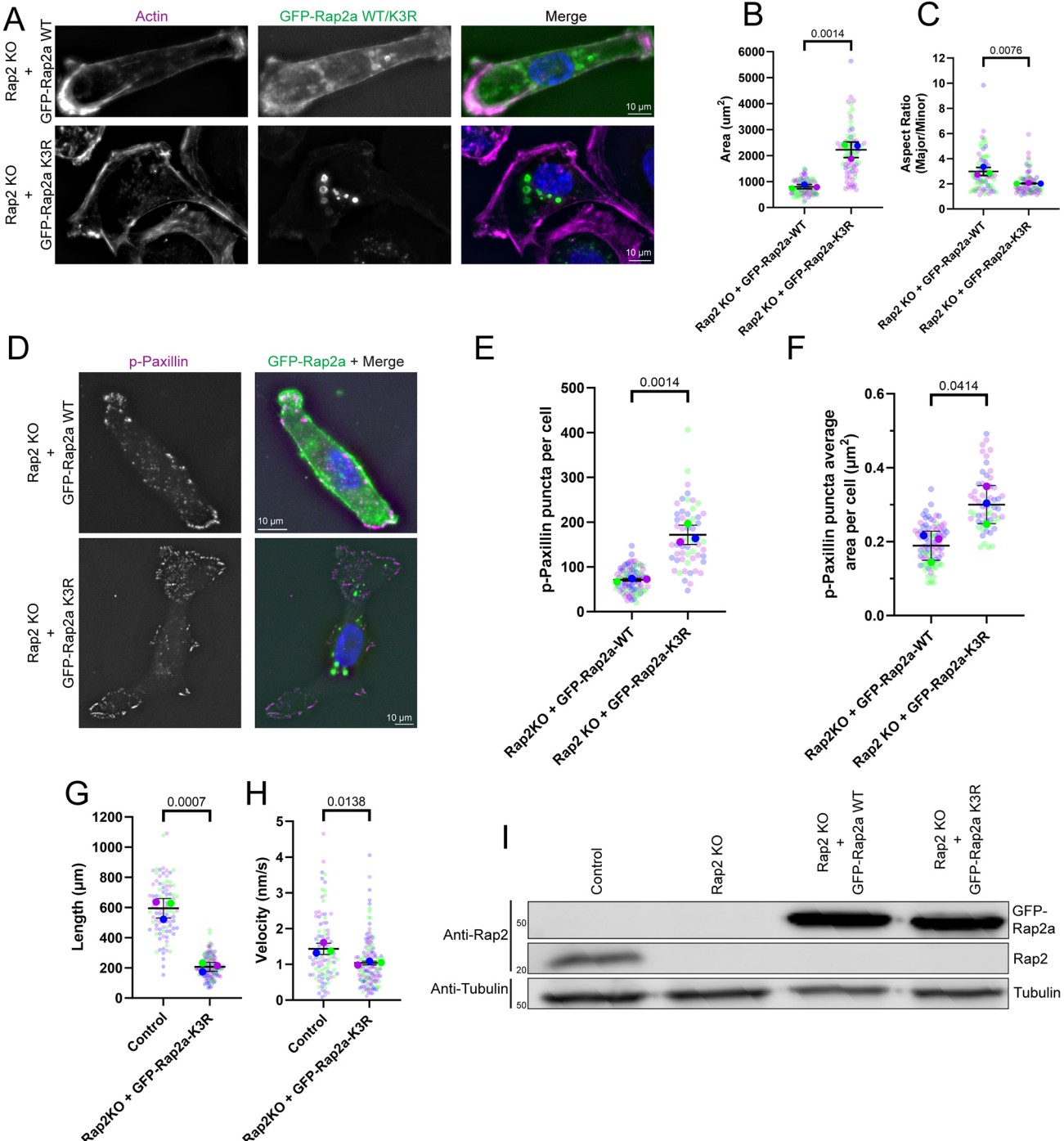

**Fig. 6. Ubiquitylation is necessary for Rap2 function during cell migration.** (A) Phalloidin staining of actin in Rap2-KO cells rescued with either a GFP–Rap2a-WT or GFP–Rap2a-K3R construct. (B,C) Quantification (*n*=3 biological replicates; control, 77 total cells; Rap2-KO, 75 total cells; average is mean of biological replicates, error bars represent biological replicate s.d.; unpaired one-tailed Student's *t*-test) of cell size (B) and cell aspect ratio (C) for cells as in A. (D) Rap2-KO cells rescued with either GFP–Rap2a-WT or GFP–Rap2a-K3R were stained with an anti-phospho-paxillin antibody (p-Paxillin). (E,F) Quantification (*n*=3 biological replicates; control, 80 total cells; Rap2-KO, 57 total cells; average is mean of biological replicates, error bars represent biological replicate s.d.; unpaired one-tailed Student's *t*-test) of FAs from cells as shown in D. (E) Average number of phospho-paxillin puncta per cell. (F) Average area of phospho-paxillin puncta per cell. (G,H) Quantification (*n*=3 biological replicates; control, 99 total cells; Rap2-KO with GFP–Rap2a-K3R, 141 total cells; average is mean of biological replicates, error bars represent biological replicate s.d.; unpaired one-tailed Student's *t*-test) of tracks from random migration assays of Rap2-KO cells rescued with either a GFP–Rap2a-WT (control) or GFP–Rap2a-K3R construct. Quantifications show the total length of the track the cells migrated (G) and the velocity at which the cells migrated (H). (I) Western blot of control, Rap2-KO and early passage Rap2-KO rescue lines (GFP–Rap2a-WT passage 3 and GFP–Rap2a-K3R passage 6), showing rescue GFP–Rap2a construct expression versus endogenous Rap2 expression. Positions of molecular mass markers are indicated in kDa. Data shown are representative of one experiment.

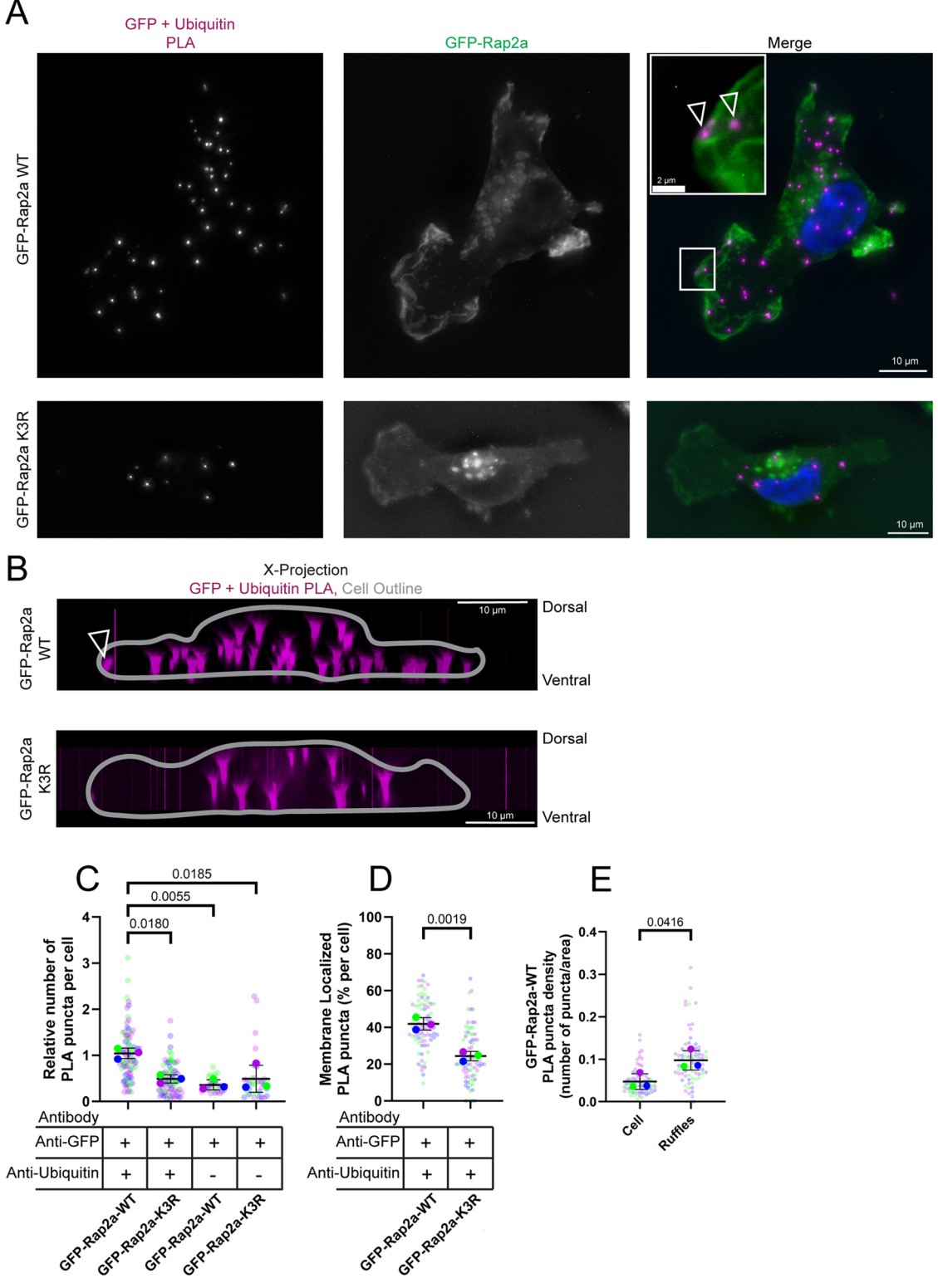

**Fig. 7. Rap2 is ubiquitylated at the plasma membrane.** (A–E) PLA of MDA-MB-231 cells expressing either GFP–Rap2a-WT or GFP–Rap2a-K3R. (A) Images of representative cells. In the zoomed-in inset image of the region marked by the box, the arrowheads highlight PLA puncta on the membrane. (B) The same images as in A projected over the image *x*-axis. The GFP–Rap2a channel was traced to show the cell outline and displayed over the PLA channel. Arrowhead marks the PLA puncta highlighted by arrowheads in A. (C) Quantification (*n*=3 biological replicates; GFP–Rap2a-WT PLA, 85 total cells; GFP–Rap2a-K3R PLA, 76 total cells; GFP–Rap2a-WT GFP control, 27 total cells; GFP–Rap2a-K3R GFP control, 32 total cells; average is mean of biological replicates, error bars represent biological replicate s.d.; one-way ANOVA with Tukey multiple comparisons, significant *P*-values shown) of PLA puncta showing the average number of PLA puncta per cell, including PLA control data from Fig. S3, all normalized per replicate to the median of the GFP–Rap2a-WT PLA data. (D,E) Quantification of (D) the localization of PLA puncta to the membrane and (E) the enrichment of PLA puncta to membrane ruffles in GFP–Rap2a-WT-expressing cells (*n*=3 biological replicates; 206 total ruffles averaged per cell, 74 total cells; average is mean of biological replicates, error bars represent biological replicate s.d.; unpaired one-tailed Student's *t*-test).

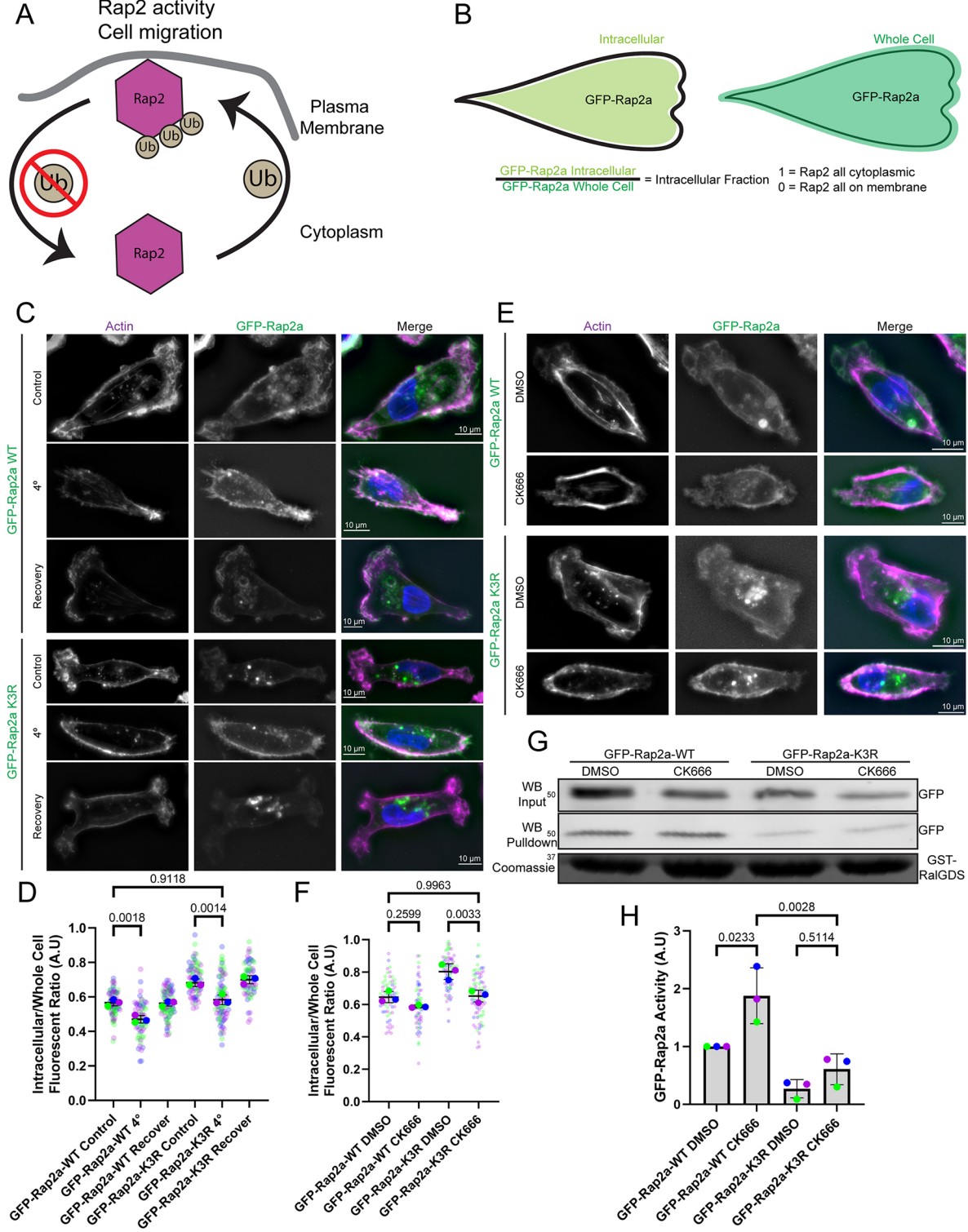

**Fig. 8.** See next page for legend.

rescue the Rap2a-K3R activation defect. To test this hypothesis, we used CK666 to induce Rap2a-K3R accumulation on the plasma membrane and tested Rap2 activity by performing Rap pulldown assays using beads bound to GST-tagged RalGDS (a known Rap2 effector protein that binds Rap2-GTP). GFP–Rap2a-WT was found to be more active when treated with CK666 (Fig. 8G,H), consistent with the idea that increased plasma membrane dwell-time increases Rap2

activation. However, CK666 treatment of cells overexpressing GFP–Rap2a-K3R only slightly increased its activation (Fig. 8G,H) indicating that increased plasma membrane retention is not sufficient to rescue the GFP–Rap2a-K3R activation defect. These results suggest that Rab40/CRL5 ubiquitylation of Rap2 likely directly regulates its activation, which as a result mediates its localization at the plasma membrane.

**Fig. 8. Ubiquitylation facilitates Rap2 retention at the plasma membrane.** (A) Schematic representation of Rap2 ubiquitylation cycle. Ub, ubiquitin. (B) Schematic representation showing the process used to calculate GFP–Rap2a plasma membrane localization. The inside and outside of the cell was traced using actin staining as a guide. (C,D) MDA-MB-231 cells expressing either GFP–Rap2a-WT or GFP–Rap2a-K3R were subject to 4°C treatment to inhibit Rap2 internalization from the plasma membrane. A 37°C temperature recovery was used to reinitiate Rap2 internalization via endocytosis and macropinocytosis. Phalloidin was used to label actin. Quantification (n=3 biological replicates; GFP–Rap2a-WT control, 86 total cells; GFP–Rap2a-WT 4°C, 66 total cells; GFP–Rap2a-WT recovery, 81 total cells; GFP–Rap2a-K3R control, 79 total cells; GFP–Rap2a-K3R 4°C, 75 total cells; GFP–Rap2a-K3R recovery, 66 total cells; average is mean of biological replicates, error bars represent biological replicate s.d.; one-way ANOVA with Tukey multiple comparisons, significant P-values shown) of Rap2 membrane localization is shown in D. (E,F) MDA-MB-231 cells expressing either GFP–Rap2a-WT or GFP–Rap2a-K3R were treated with CK666 to inhibit macropinocytosis-dependent internalization of Rap2 or with DMSO as a control. Phalloidin was used to label actin. Quantification (n=3 biological replicates; GFP–Rap2a-WT DMSO, 65 total cells; GFP–Rap2a-WT CK666, 63 total cells; GFP–Rap2a-K3R DMSO, 73 total cells; GFP–Rap2a-K3R CK666, 78 total cells; average is mean of biological replicates, error bars represent biological replicate s.d.; one-way ANOVA with Tukey multiple comparisons, significant P-values shown) of Rap2 membrane localization is shown in F. (G,H) GST–RalGDS bead pulldown assays were used to test the levels of activated GFP–Rap2a-WT and GFP–Rap2a-K3R after CK666 treatment. Rap2 binding to GST–RalGDS was assessed by western blot (WB), and Coomassie Blue staining was used to confirm the loading of GST–RalGDS into cell lysates. Quantification (n=3 biological replicates; each replicate normalized to GFP–Rap2a-WT DMSO pulldown value; average is mean of biological replicates, error bars represent biological replicate s.d.; one-way ANOVA with Tukey multiple comparisons, significant P-values of vital comparisons shown) of western blots of GFP–Rap2a binding to GST–RalGDS normalized to GFP–Rap2a levels in the lysate is shown in H. Positions of molecular mass markers are indicated in kDa. A.U., arbitrary units.

## Ubiquitylation regulates GEF-dependent Rap2 activation

Our data suggest that inhibition of Rab40/CRL5-dependent ubiquitylation inhibits Rap2 activation (GTP binding) and decreases Rap2 dwell-time at the membrane. Furthermore, we show that although we can rescue Rap2a-K3R targeting to the plasma membrane by inhibiting macropinocytosis, it does not rescue defects in Rap2a-K3R activation. These data indicate that ubiquitylation dependent GTP loading of Rap2 is what maintains Rap2 localization at the membrane. To further test this hypothesis, we treated cells overexpressing GFP–Rap2a-WT or GFP–Rap2a-K3R with 8-bromo-cAMP, a cell permeable cAMP analog (Wang and Adjaye, 2011; Enserink et al., 2002). cAMP stimulates activation of EPAC1 (also known as RapGEF3), EPAC2 (also known as RapGEF4) and RapGEF2, well known plasma membrane-associated GEFs for Rap2 (Kumar et al., 2018; Sartre et al., 2023; Kuiperij et al., 2003). As expected, we show that 8-bromo-cAMP treatment resulted in an increase of GFP–Rap2a-WT on the plasma membrane (Fig. 9A,B). Although the GFP–Rap2a-WT increase on the plasma membrane was only moderate, it is regarded that the majority of the Rap2 population exists in an active state (Liu et al., 2010; Ohba et al., 2000). Thus, only a small proportion of Rap2 is available to respond to increased GEF activity. As such, the moderate increase of GFP–Rap2a-WT on the membrane is still consistent with the hypothesis that activation is what maintains Rap2 at the plasma membrane. Furthermore, the increased GFP–Rap2a-WT on the membrane was asymmetric, with distinct enrichment noticed at areas of actin ruffling, thus supporting the idea of active Rap2 being localized to the lamellipodium where it functions as a RhoA inhibitor at the lamellipodium (Fig. 9A). Importantly, 8-bromo-cAMP

treatment did not change cytoplasmic localization of GFP–Rap2a-K3R (Fig. 9A,B), indicating that ubiquitylation is needed for Rap2 to respond to its cAMP responsive GEFs.

We next decided to further analyze the role of Rap2 activity by generating constitutively active Rap2 mutants. The Rap GTPase family member Rap1 has been shown to be constitutively activate upon a Q63E mutation (Zhang et al., 2014; Dao et al., 2009). Similar Q-to-E mutations have also been shown to generate constitutively active forms of other small monomeric GTPases, such as Ras and Rab GTPases. (Langemeyer et al., 2014; Gopal Krishnan et al., 2020; Prior et al., 2012). Thus, we made the Q63E mutation in Rap2 to create a GFP–Rap2a-Q63E mutant and performed GST–RalGDS pulldown assays to test the activity of the Rap2-Q63E mutant. As shown in Fig. 9C,D, GFP–Rap2a-Q63E bound GST–RalGDS beads stronger than GFP–Rap2a-WT, showing that Rap2-Q63E is a constitutively active mutant. Accordingly, we tested the effect of K117, K148 and K150 ubiquitylation on activation of Rap2-Q63E by creating a GFP–Rap2a-Q63E-K3R mutant. Using GST–RalGDS pulldown assays we found that, like the GFP–Rap2a-K3R mutant, GFP–Rap2a-Q63E-K3R is largely inactive (Fig. 9E,F). It has been proposed that Q mutations in the switch-II region of Ras GTPases block their interaction with GAPs (Vetter and Wittinghofer, 2001; Prior et al., 2012), thus locking them in the GTP-bound state. As such, the Rap2-Q63E mutant is likely unable to be deactivated by GAPs, promoting its presence in the GTP-bound state. Since the combination of the Rap2-Q63E-K3R mutations renders Rap2 inactive, the ubiquitylation of Rap2 must promote its ability to be activated, likely by facilitating Rap2 interaction with GEFs. As such, using these mutations, we were able to dissect how activation by GEFs and deactivation by GAPs regulates Rap2 localization.

To further confirm that ubiquitylation is necessary for Rap2 activation, we treated cells expressing GFP–Rap2a-Q63E-K3R with CK666 to force the proximity of Rap2 with membrane-localized GEFs, and performed GST–RalGDS pulldowns. Whereas decreasing internalization from the plasma membrane through CK666 treatment increases the activation of GFP–Rap2a-WT, likely as a result of increasing Rap2 localization with its GEFs (Fig. 8G,H), CK666 treatment did not increase GFP–Rap2a-Q63E-K3R activation (Fig. 9G,H). Since inhibiting the association of Rap2-K3R with its GAPs (by making the Q63E mutation) and increasing its presence at the plasma membrane where Rap2 GEFs are localized (CK666 treatment) did not rescue Rap2-K3R activation defects, it is likely that Rab40/CRL5-dependent ubiquitylation is required for Rap2 to be activated by its cognate GEF.

Next, using our Q63E mutants with decreased deactivation by GAPs, we sought to understand whether ubiquitylation-dependent Rap2 activation by GEFs regulates Rap2 localization. As such, we observed the localization of our GFP–Rap2a-Q63E and GFP–Rap2a-Q63E-K3R constructs in MDA-MB-231 cells. GFP–Rap2a-Q63E was observed to localize more strongly to the lamellipodium and plasma membrane than GFP–Rap2a-WT (Fig. 9I,J), suggesting that increased activity as a result of decreased Rap2–GAP interaction increases Rap2 localization at the plasma membrane, specifically at the lamellipodium. Interestingly, the Q63E mutation did not rescue localization of the GFP–Rap2a-Q63E-K3R (Fig. 9I,J), further supporting the hypothesis that ubiquitylation-dependent Rap2 activation by GEFs is necessary for Rap2 localization at the plasma membrane.

Altogether, our data suggest that Rap2 activation regulates its localization by increasing Rap2 dwell-time at the lamellipodium membrane. Accordingly, it is predicted that CK666-induced accumulation of wild-type Rap2 on the membrane should be

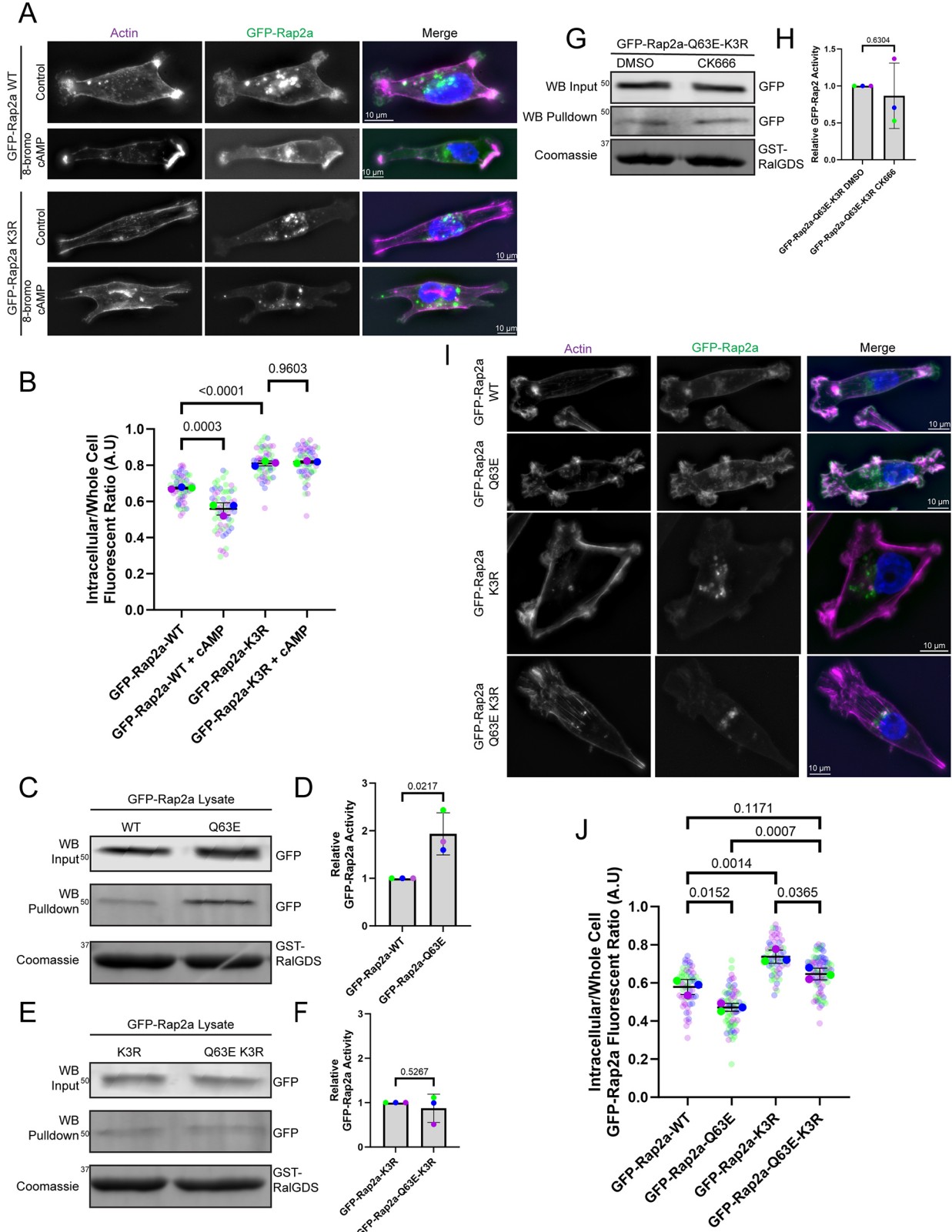

**Fig. 9.** See next page for legend.

maintained after CK666 removal, whereas the Rap2-K3R mutant should be removed from the membrane as it was never able to be activated. In accordance with these predictions, washout of CK666 resulted in rapid GFP–Rap2a-K3R removal from

the plasma membrane, whereas GFP–Rap2a-WT remained at the lamellipodium membrane at similar levels as during CK666 treatment (Fig. S4A,B). These results display how increased GFP–Rap2a-WT activation increases Rap2 localization at the

**Fig. 9. Ubiquitylation mediates GEF-dependent activation of Rap2.**
(A,B) 8-bromo-cAMP was added to MDA-MB-231 cells expressing either GFP–Rap2a-WT or GFP–Rap2a-K3R in order to increase activity of a plasma membrane-associated Rap GEF and, accordingly, increase Rap2 activity. Phalloidin was used to label actin. Quantification (n=3 biological replicates; GFP–Rap2a-WT, 57 total cells; GFP–Rap2a-WT 8-bromo-cAMP, 61 total cells; GFP–Rap2a-K3R, 53 total cells; GFP–Rap2a-K3R 8-bromo-cAMP, 56 total cells; average is mean of biological replicates, error bars represent biological replicate s.d.; one-way ANOVA with Tukey multiple comparisons, significant $P$-values of vital comparisons shown) of Rap2 membrane localization is shown in B. (C–F) GST–RalGDS bead pulldown assays were used to measure the levels of activated GFP–Rap2a-WT and GFP–Rap2a-Q63E (C), and GFP–Rap2a-K3R and GFP–Rap2a-Q63E-K3R (E). Quantification (n=3 biological replicates; each replicate averaged to GFP–Rap2a-WT or GFP–Rap2a-K3R pulldown value; average is mean of biological replicates, error bars represent biological replicate s.d.; unpaired one-tailed Student's $t$-test) of GFP–Rap2a binding to GST–RalGDS beads was normalized to GFP–Rap2a levels in the lysate (D,F). (G,H) GST–RalGDS bead pulldowns were used to assess how CK666 affects GFP–Rap2a-Q63E-K3R activity. Quantification (n=3 biological replicates; each replicate averaged to GFP–Rap2a-Q63E-K3R DMSO pulldown value; average is mean of biological replicates, error bars represent biological replicate s.d.; unpaired one-tailed Student's $t$-test) of GFP–Rap2a binding to GST–RalGDS beads was normalized to GFP–Rap2a levels in the lysate (H). (I,J) Localization of constitutively active GFP–Rap2a constructs in MDA-MB-231 cells. Phalloidin was used to label actin. Quantification (n=3 biological replicates; GFP–Rap2a-WT, 64 total cells; GFP–Rap2a-Q63E, 70 total cells; GFP–Rap2a-K3R, 63 total cells; GFP–Rap2a-Q63E-K3R, 77 total cells; average is mean of biological replicates, error bars represent biological replicate s.d.; one-way ANOVA with Tukey multiple comparisons, significant $P$-values shown) of Rap2 membrane localization is shown in J. In C,E,G, positions of molecular mass markers are indicated in kDa. A.U., arbitrary units

lamellipodium plasma membrane and how Rab40/CRL5-dependent ubiquitylation is necessary for GEF-dependent Rap2 activity.

## DISCUSSION
Cell migration is a vital process relying on the precise coordination of many intracellular molecular mechanisms to establish front-to-back polarity and create functional migratory structures. Here, we describe the mechanism by which Rab40/CRL5-dependent ubiquitylation regulates the spatiotemporal dynamics of Rap2. We also demonstrate that Rap2 functions as a RhoA inhibitor that likely functions by recruiting ARHGAP29 during retraction of lamellipodia ruffles, thus promoting lamellipodia dynamics and cell migration.

### The Rap2 GTPase as a major facilitator of lamellipodia dynamics
Mesenchymal cell migration is hallmarked by the formation of a lamellipodium at the leading edge of the cell. Canonically, the lamellipodium defines the direction of cell migration and forms as a result of polarized Rac1 activation at the leading edge of the cell. Rac1 then activates the Arp2/3 complex, which nucleates branched actin polymerization, creating the pushing force at the front of the lamellipodium membrane that drives membrane extension (Suraneni et al., 2012; Bisi et al., 2013; Schaks et al., 2019). However, recently it has been shown that RhoA activity is also needed at the lamellipodium to facilitate ruffling and allow for lamellipodia dynamic extension and retraction cycles (O'Connor et al., 2000; O'Connor and Chen, 2013; Kurokawa and Matsuda, 2005). We show that Rap2 regulates lamellipodia dynamics by acting as a negative regulator of RhoA. Specifically, we propose that Rap2 recruitment to retracting lamellipodia ruffles inhibits RhoA-induced acto-myosin contraction, terminating ruffle retraction and promoting branched actin polymerization and lamellipodia

extension (Fig. 10A). Consistent with this hypothesis, Rap2 knockout increases ruffle retraction distance and decreases ruffling periodicity. This decrease in lamellipodia ruffling likely contributes to the inhibition of cell migration in Rap2-KO cells.

Our study also shows that the effect of Rap2 loss was not limited to defects in lamellipodia dynamics. Specifically, FAs were seen to be more stable and acto-myosin stress fibers were more prevalent and contractile in Rap2-KO cells. Importantly, the connection between FAs and stress fibers is well established as stress fibers anchor to FAs. Furthermore, increased stress fiber contractility leads to activation and stabilization of FAs (Burridge and Guilluy, 2016; Bachmann et al., 2022; Wolfenson et al., 2011). Finally, regulation of FA assembly–disassembly dynamics is known to play a key role in regulating cell migration (Wehrle-Haller, 2012; Mishra and Manavathi, 2021; Chen, 1979). While Rap2 can regulate integrin trafficking in T-cells and thyroid cancer cells (Stanley et al., 2012; Miertzschke et al., 2007; Dong et al., 2012), both FAs and acto-myosin stress fibers are also structures regulated by RhoA signaling (Hotulainen and Lappalainen, 2006; Julian and Olson, 2014; Chrzanowska-Wodnicka and Burridge, 1996; Ridley and Hall, 1992). Thus, Rap2 regulated RhoA activity is likely necessary for maintaining FAs and stress fibers. Defects in all these structures work together as strong driving factors to explain the loss of migration observed in Rap2-KO cells. However, future work is needed to dissect the differences in phenotypes that result from Rap2 trafficking of FAs and Rap2 regulation of RhoA activity.

### Rap2 as a regulator of ARGHAP29
The spatiotemporal dynamics of RhoA are regulated by RhoA GEFs and GAPs, with GAPs mediating the generation of GDP-bound (inactive) RhoA. Notably, many RhoA GAPs are known to exist, multiple of which have been implicated in regulating cell migration (Mosaddeghzadeh and Ahmadian, 2021). The specific functions of many of these GAPs, and which GAPs mediate RhoA inhibition at the lamellipodium, remain to be understood. Here, we demonstrate that Rap2 regulation of RhoA is, at least in part, through interaction with ARHGAP29 at the lamellipodium membrane. Interestingly, while this interaction explains the changes in lamellipodia dynamics upon Rap2 knockout, the lack of ARHGAP29 enrichment at stress fibers or FAs leaves the question of whether other RhoA GAPs can also be recruited by Rap2 to the FAs and stress fibers. It is also possible that increased RhoA activity and signaling at the lamellipodium can propagate across the whole cell and stimulate RhoA-regulated structures. Further work is required to determine whether Rap2 can also bind and regulate other Rho GAPs.

Interestingly, the interaction between ARHGAP29 and Rap2 was found to require the presence of membranes. We speculate that this is a result of the ARHGAP29 C1 domain binding to DAG and F-BAR domain binding to curved membranes at the lamellipodium, causing a structural shift that allows for Rap2 interaction. The idea of F-BAR domains as being autoinhibitory of protein function has been previously established in other migration-related proteins (Stanishneva-Konovalova et al., 2016; de Kreuk et al., 2013). In fact, ARHGAP45 (HMHA1), a structurally similar GAP to ARHGAP29 (Mosaddeghzadeh and Ahmadian, 2021), has been shown to have its RhoA GAP activity autoinhibited by its F-BAR domain. Membrane binding releases the RhoGAP domain, allowing for ARHGAP45 function (de Kreuk et al., 2013). Future work will be needed to establish whether a similar autoinhibition by the F-BAR domain in ARHGAP29 also regulates Rap2 binding and recruitment to the leading edge of migrating cells.

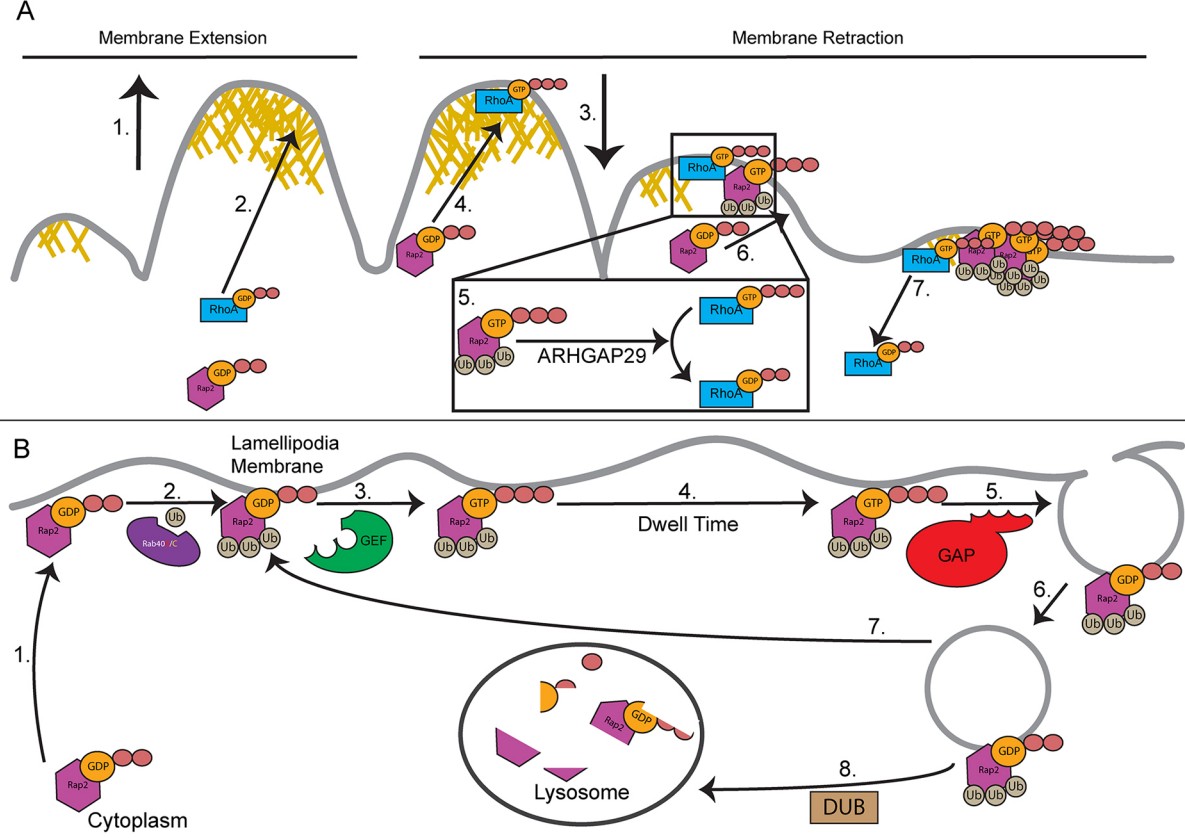

**Fig. 10. Ubiquitylation mediates Rap2 activation, localization and function during cell migration.** (A) Proposed model of Rap2 function during cell migration. Membrane extension at the lamellipodium is hallmarked by branched actin polymerization, pushing the membrane outwards (1). At the end of the extension event, RhoA is recruited to the membrane and is activated (2), where it functions to mediate ruffle retraction (3). As the ruffle retracts, Rap2 is recruited to the retracting ruffle (4). Once localized, Rap2 is activated and inhibits RhoA activity, through interactions with ARHGAP29 (5). Continued Rap2 recruitment (6) increases Rap2 inhibition of RhoA, limiting the extent of ruffle retraction and removing RhoA activity from the membrane (7). Rap2 will then be deactivated and removed from the membrane allowing for another ruffle extension–retraction cycle. (B) Proposed model of the role of Rab40/CRL5 ubiquitylation in regulating Rap2 localization and activation at the lamellipodium membrane. Rap2 is trafficked from the cytoplasm to the lamellipodium membrane (1). On the membrane, it is met and ubiquitylated by the Rab40/CRL5 E3 ubiquitin ligase complex (2). Once ubiquitylated, Rap2 is activated by a GEF (3). Active (GTP-bound) Rap2 is retained at the plasma membrane, where it functions to regulate lamellipodia dynamics (4). After Rap2 is inactivated by GAPs (5), it is internalized through macropinocytosis (6). When internalized into an early endosome, Rap2 can either be recycled back to the plasma membrane, where it is reactivated (7) or trafficked to the lysosome for degradation (8). We speculate that the decision between Rap2 recycling or trafficking to the lysosome is based on whether Rap2 is deubiquitylated by a deubiquitylase, leading to lysosomal degradation, or whether it maintains its ubiquitylation and is recycled. DUB, deubiquitylase; Ub, ubiquitin.

Here we propose a role for Rap2 in regulating ruffle retraction through ARHGAP29 inhibition of RhoA. Importantly, RhoA activity at the lamellipodium has been shown to be involved in leading edge membrane protrusions (Jacquemet et al., 2013; Pertz et al., 2006; Heasman and Ridley, 2010). Specifically, RhoA activation of mDia (also known as DIAPH1), a formin that polymerizes linear actin filaments, has been shown to be necessary for leading edge membrane ruffling and extension and for maintaining cell polarity (Kurokawa and Matsuda, 2005; Vicente-Manzanares et al., 2003). While exploring how Rap2 and ARHGAP29 regulate mDia falls beyond the scope of this work, it remains interesting to speculate how specific regulation of RhoA regulates its role in multiple leading edge functions. It has been established that loss of GEF-H1 (also known as ARHGEF2), a RhoA GEF, does not alter the levels of RhoA activity, but instead alters the spatiotemporal dynamics of RhoA activity at the leading edge (Nalbant et al., 2009), suggesting that any disruption of RhoA regulation may have large effects on downstream RhoA signaling. As we show that loss of Rap2 increases the temporal activity of RhoA (Fig. 3B–D) and perturbs the lamellipodium as a whole (Fig. 1D–J), it is reasonable to assume that Rap2 loss indirectly alters RhoA regulation of mDia at the leading edge. However,

identification of whether this is as a direct result of Rap2 interaction with ARHGAP29 or as a result of large-scale changes in RhoA activity remains an outstanding question.

### Ubiquitylation mediates Rap2 targeting to the lamellipodia membrane

Rab40/CRL5 E3 ubiquitin ligase-dependent ubiquitylation of Rap2 has been shown to be necessary for its localization to the plasma membrane and lamellipodia ruffles (Duncan et al., 2022). However, it remained unknown where and when Rap2 is ubiquitylated and how this ubiquitylation mediates Rap2 targeting to the lamellipodium. In this study we used PLA to suggest that Rap2 is ubiquitylated at the plasma membrane and that ubiquitylated Rap2 is specifically enriched at lamellipodia ruffles. This observation is consistent with our previous study showing that the Rab40/CRL5 complex is also enriched at the lamellipodium ruffles (Duncan et al., 2022; Linklater et al., 2021). In line with these results, we propose that, upon delivery to lamellipodia plasma membrane, Rap2 is ubiquitylated by the Rab40/CRL5 complex. This ubiquitylation is required for Rap2 to accumulate and function at the lamellipodium. Interestingly, some ubiquitylated Rap2 can

also be observed at endocytic vesicles. This observation suggests that in addition to controlling Rap2 localization to the membrane, ubiquitylation of Rap2 might also regulate Rap2 function beyond the plasma membrane and lamellipodium. Further work, however, will be necessary to conclusively answer these questions.

The observation that ubiquitylated Rap2 is enriched at the lamellipodium membrane raises the question of how ubiquitylation mechanistically mediates Rap2 targeting to the plasma membrane. Canonically, ubiquitylation is a signal for plasma membrane-associated proteins, such as tyrosine kinase receptors, to be internalized and trafficked to the lysosome for degradation (Weissman et al., 2011; Critchley et al., 2018; Tanner et al., 2019; Clague and Urbé, 2017). However, ubiquitylation has also been shown to regulate membrane protein trafficking without promoting degradation (Date et al., 2022; Xu et al., 2017; Choi et al., 2018). Although ubiquitylation does not lead to Rap2 degradation, these past works supported the idea that ubiquitylation can be directly involved in regulating protein delivery to or removal from the plasma membrane. Our data suggest that ubiquitylation does not directly regulate Rap2 trafficking to the plasma membrane, as defects in Rap2-K3R localization can be rescued by inhibiting Arp2/3-dependent lamellipodia ruffling that drives macropinocytosis-dependent internalization of Rap2 (Duncan et al., 2022). Furthermore, re-initiation of lamellipodia ruffling by washing out the Arp2/3 inhibitor leads to rapid removal of Rap2-K3R from plasma membrane and its accumulation in endosomes and lysosomes. These results suggest that ubiquitylation increases Rap2 dwell-time at the lamellipodium membrane by inhibiting its removal from the plasma membrane. However, more work is needed to understand the molecular mechanism by which ubiquitylation might govern Rap2 internalization. Furthermore, how Rap2 is deubiquitylated, and the resulting consequence of this deubiquitylation on Rap2 trafficking, still needs to be determined.

## Ubiquitylation is necessary for GEF-dependent Rap2 activation

Small GTPases are canonically thought of as being activated by GEFs. For Ras and Rap GTPases, activation is driven by plasma membrane-associated GEFs. Active (GTP-bound) Rap GTPases are then maintained at the plasma membrane, where they function to mediate signaling and localized cytoskeleton dynamics. Eventual inactivation by GAPs results in internalization of GTPases and their removal from the plasma membrane (Bos et al., 2007; Segev, 2011). Consistent with this idea, we showed that the overactivation of a Rap2 GEF increases Rap2 plasma membrane localization in a ubiquitylation-dependent manner. We further showed that enhanced localization to the plasma membrane increases Rap2 activity, likely due to increased colocalization with its membrane-localized GEFs. In contrast to Rap2-WT, increasing Rap2-K3R localization to the plasma membrane only slightly increased its activation. Furthermore, using cAMP to activate EPAC1, EPAC2 and RapGEF2, known plasma membrane-associated GEFs for Rap2, did not rescue the activation defects of Rap2-K3R mutants. Thus, this suggests that ubiquitylation of Rap2 might directly modulate its ability to bind and be activated by its cognate GEF. Notably, the identification of a GEF for which ubiquitylation facilitates Rap2 interaction is beyond the scope of this work. Five major Rap2 GEFs have been previously identified: EPAC1, EPAC2, RapGEF2, RasGEF1, and C3G (also known as RapGEF1) (Radha et al., 2011; de Rooij et al., 2000; Yaman et al., 2009; Kuiperij et al., 2003). Future work is necessary to assess GEF activity of these proteins on both Rap2a-WT and Rap2a-K3R.

To further explore the requirement for ubiquitylation in Rap2 activation, we generated Rap2-Q63E and Rap2-Q63E-K3R constitutively active Rap2 mutants. The Q63E mutation has been shown to be constitutively active in Rap1 (Zhang et al., 2014; Dao et al., 2009), and equivalent mutations are constitutively active in other small monomeric GTPases (Langemeyer et al., 2014; Gopal Krishnan et al., 2020; Prior et al., 2012; Vetter and Wittinghofer, 2001). In Ras and Rap GTPases, the conserved Q residues are located in the switch-II region and line the entrance to the GTP binding pocket, where a GAP needs to insert its arginine finger to facilitate the GTP-to-GDP conversion reaction. Consequently, mutation of these Q residues hinders Ras and Rap interactions with their cognate GAPs (Prior et al., 2012; Vetter and Wittinghofer, 2001). Consistent with this, we show that the Rap2-Q63E mutation leads to activation and enhanced targeting of Rap2 to the plasma membrane. Interestingly, the Rap2-Q63E-K3R mutant failed to localize to the plasma membrane. Furthermore, the levels of active Rap2-Q63E-K3R were found to be similar to those of Rap2-K3R, suggesting that ubiquitylation is necessary for Rap2 activation, potentially through directly regulating Rap2 interactions with its GEF.

Based on these results, we propose a Rab40/CRL5-dependent mechanism that regulates Rap2 activity during cell migration (Fig. 10A). Once Rap2 is delivered to the lamellipodium membrane (Duncan et al., 2022), it is ubiquitylated by a Rab40/CRL5 complex that is enriched at the lamellipodium (Duncan et al., 2022; Linklater et al., 2021). This ubiquitylation mediates Rap2 interaction and activation by its cognate GEF. Active Rap2 is then retained at the plasma membrane where it functions to regulate lamellipodia dynamics during cell migration (Fig. 10B). It remains to be determined whether the sole purpose of Rap2 ubiquitylation is to allow for its activation or if ubiquitylation also promotes Rap2 binding with its effectors. In other words, the question remains, if a Rap2a-K3R mutant is forced to be in its GTP-bound state, can it still bind its effectors? While we propose the Rap2–ARHGAP29 interaction, Rap2 effectors β-arrestin 1 and the Rassf family member RAPL (encoded by RASSF5) (Miertzschke et al., 2007; Gera et al., 2017) have also been shown to interact with Rap2. The role of ubiquitylation in these, and all future Rap2 effectors, remains to be evaluated.

Notably, the Ras GTPase family member KRAS has been shown to be mono-ubiquitylated at K147. It has also been demonstrated that mono-ubiquitylation of KRAS at K147 inhibits its interaction with its GAP, thus increasing KRAS activation (Choi et al., 2018; Sasaki et al., 2011). Furthermore, the Ras GTPase family member Rap1 has also been shown to be ubiquitylated (likely mono-ubiquitylated) at multiple residues as a way to control its activity. Ubiquitylation at K31 switches Rap1 affinity towards different effector complexes, fine-tuning its activity (Sewduth et al., 2023). Additionally, we have also shown that Rab40/CRL5 ubiquitylation of Rap1b drives its activation, though interestingly, this ubiquitylation promotes Rap1b localization away from the leading edge. However, whether ubiquitylation also regulates GEF-dependent activation of Rap1b remains unknown (Neumann et al., 2025). Finally, Rac1 has also been shown to be ubiquitylated by the E3 ubiquitin ligase TRAF6, resulting in increased Rac1 activity (Li et al., 2017). While we show that Rap2 ubiquitylation functions to allow for GEF-dependent activation, it is intriguing that multiple small GTPases involved in regulating cell migration have evolved mechanisms of mono-ubiquitylation-dependent activation.

This conserved mechanism presents an exciting area of future cell migration research. As major regulators of cell migration and actin dynamics, understanding how ubiquitylation facilitates localization

and activation of Ras, Rap, and Rac GTPases will lead to new insights into fine-tuning cell migration as well as providing new modifications to target for treating cell migration-related disorders and diseases.

## Study limitations

This study suggests that Rap2 ubiquitylation by the Rab40/CRL5 complex regulates GEF-dependent Rap2 activation, which in turn regulates its localization. However, additional experiments beyond the scope of this work could be performed to further support this claim. First, we still do not know what GEF regulates Rap2 activation during cell migration. Identification of the GEF will be needed to demonstrate that Rap2 ubiquitylation does directly affect GEF binding to Rap2. Second, while experimental evidence suggests that Rap2 is mono-ubiquitylated on K117, K148 and/or K150 in a Rab40/CRL5-dependent fashion (Duncan et al., 2022), this remains to be unequivocally demonstrated. Thus, this study cannot fully rule out the possibility that the K3R mutation also affects other aspects of Rap2 function that are not dependent on ubiquitylation.

## MATERIALS AND METHODS

### Cell culture

MDA-MB-231 cells (ATCC) were cultured in 231 medium [DMEM with 4.5 g/l glucose, 5.84 g/l L-glutamine (Corning), 1% sodium pyruvate (Gibco), 1% nonessential amino acids (Gibco), 1 µg/ml insulin (Gibco), 1% penicillin-streptomycin (Corning), and 10% fetal bovine serum (FBS; Phoenix Scientific)]. HEK293T cells (ATCC) were cultured in 293T medium [DMEM with 4.5 g/l glucose, 5.84 g/l L-glutamine (Corning), 1% penicillin-streptomycin (Corning), and 10% FBS (Phoenix Scientific)]. All MDA-MB-231 stable cell lines used in this study were generated using lentivirus plasmid pLVX as described previously (Duncan et al., 2022). The Rap2-KO line had been previously created and validated (Duncan et al., 2022). Cell lines were routinely tested for mycoplasma. Additionally, all cell lines were authenticated in accordance with ATCC standards.

### Generation of lentiviral stable cell lines

Calcium phosphate was used to transfect HEK293T cells (50% confluent) with pLVX plasmid containing GFP-tagged genes of interest (Rap2a constructs). After 6 h the medium was replaced with fresh 293T medium. Cells were left for 48 h to allow for the virus to accumulate in the medium. Viral 293T medium was collected, filtered through a 0.45 µm PVDF low-binding syringe filter, and treated with polybrene (Millipore Sigma; 100 µg per 1 ml medium). Viral 293T medium was added to target MDA-MB-231 cells (50% confluent) for 2 h. Medium was replaced with 231 medium, and target cells were allowed to recover for 24 h, then selected with puromycin (5 µg/ml). Expression for the GFP-tagged gene of interest in cell lines was then validated using western blotting. Ubiquitylation mutant Rap2a proteins degrade quickly and do not express at the same level as wild-type Rap2a constructs (Duncan et al., 2022). As such, experiments using mutant Rap2a were performed using low-passage cells. Furthermore, as passaging continued, data were compared to lower passage number data to ensure that decreasing expression did not affect experimental outcomes. Furthermore, stable cell lines were remade when Rap2a construct fluorescence became too dim for use.

### Transient transfection of MDA-MB-231 cells

Transient transfections were carried out using the X-tremeGENE transfection reagent following the manufacturer's recommended protocol. For transfection with GFP-paxillin (Addgene plasmid 50529), cells were seeded in 35 mm glass-bottom dishes and transfected the same day. Time-lapse imaging was performed the following day. For the transfection with the RhoA biosensor (dTom-2×rGBD; Mahlandt el al., 2021), cells were plated in a 35 mm glass-bottom dish like before, except transfection was performed the next day. Time-lapse imaging was performed the day after transfection.

### Random migration assay

A single-cell migration assay was performed using live-cell imaging. Imaging was conducted on an OLYMPUS IX83 inverted confocal microscope with 10× air objective and a temperature-controlled stage top, which was maintained at 37°C with 5% $CO_2$. Cells were plated uniformly in a 35 mm glass-bottom Petri dish that was pre-coated with fibronectin (Millipore Sigma). Fibronectin coating was performed by allowing the fibronectin solution to dry for 1 h under UV light. Cells were then seeded and allowed to adhere overnight. The next day, the old culture medium was replaced with a fresh medium the next day before starting the live-cell imaging experiment. The time-lapse imaging was automatically taken and set to take at 20 min intervals, capturing a total of 36 frames, resulting in a 12 h time-lapse movie.

### Immunofluorescence staining

MDA-MB- 231 cells were seeded onto 1× collagen-coated glass cover glass slips (rat tail collagen, isolated in-house) and grown in full growth medium for ~24 h. Cells were later washed with room temperature PBS (TBS in case of phospho-antibody staining) and fixed it with 4% paraformaldehyde for 15 min at room temperature. Cells were then quenched for 5 min with Quench buffer (375 mg of glycine diluted in PBS) and incubated in Incubation buffer (1 ml of FBS, PBS, 1% bovine serum albumin, 200 mg saponin) for 30 min. Acti-stain 555 (phalloidin) was diluted at 1:100 and incubated with cells for 30 min in a humidified chamber. Cells were treated with 1:1600–1:2000 Hoechst 33342 stain for 5 min, then washed twice with PBS. Coverslips were then mounted onto glass slides using Vectashield (Vector Labs).

### Stroboscopic analysis of cell dynamics

SACED was performed as previously described (Hinz et al., 1999). Cells were plated in collagen-coated glass-bottom dishes and allowed to adhere overnight. Then, 1 h before imaging, 231 medium was buffered with 40 mM Hepes. Imaging was performed using a 64× differential interference contrast (DIC) objective. Control cells were selected for imaging if they exhibit distinct ruffling lamellipodia and defined polarity. Due to the loss of polarity and lamellipodia in Rap2-KO cells, these cells were selected based on the presence of a ruffling lamellipodia-like structure. Before image acquisition, the field of view was cropped and focused to the ruffling edges of the cells. Images were taken for 8 min, at 1 frame a second, with an exposure time of 100 ms. This was repeated across three replicates with multiple cells per replicate (Fig. 2A–C). Only cells that maintained focus of the ruffling edge throughout the whole 8 min video were selected for analysis.

### Cell lysis and western blotting

Unless otherwise stated, cells were lysed on ice in buffer containing 20 mM Hepes, pH 7.4, 150 mM NaCl, 1% Triton X-100 and 1 mM PMSF. After 30 min, lysates were clarified at 15,000 $g$ in a prechilled microcentrifuge. Supernatants were collected and analyzed via Bradford assay. Lysate samples were prepared in 5× SDS loading dye [300 mM Tris-HCl, pH 6.8 (RPI), 0.05% Bromophenol Blue (Fisher Biotech), 50% glycerol (Thermo Fisher Scientific), 10%SDS (RPI), 2-mercaptoethanol (Millipore Sigma)], boiled for 5 min at 95°C, and separated via SDS-PAGE. Gels were transferred onto 0.45 µm polyvinylidene difluoride membrane, followed by blocking for 1 h in Intercept Blocking Buffer diluted 1:3 in TBS containing 0.05% Tween (TBST). Primary antibodies (made in diluted Intercept Blocking Buffer) (Table S1) were incubated overnight at 4°C. Blots were then washed in TBST followed by incubation with IRDye fluorescent secondary antibody (in diluted Intercept Blocking Buffer) for 1 h at room temperature. Blots were washed once again with TBST before final imaging on a Li-Cor Odyssey CLx. Original blots are shown in Figs S5, S6 and S7. Details of antibodies are shown in Table S1.

### GFP-Rap2a and ARHGAP29 binding assay

GST-GFP-Trap was purified as described previously for GST-RalGDS (Duncan et al., 2022). GFP-Rap2a overexpressing MDA-MB-231 cells were grown in a 10 cm dish to 80% confluency. For homogenized sample pulldowns, plates were rinsed once in room temperature buffer containing 20 mM Hepes pH 7.4 and 150 mM NaCl then incubated at room temperature in 5 ml of 10 mM Hepes. Cells were then scraped into 200 µl

of 10 mM Hepes and pooled in a Dounce homogenizer. Cells were burst with 20 strokes of the homogenizer. Burst cells were then spun at 1000 ***g*** for 5 min in a cold centrifuge to pellet nuclei, and the supernatant was taken and brought to 20 mM Hepes, pH 7.4, 150 mM NaCl, 1 mM PMSF and 5 mM iodoacetamide [deubiquitylase (DUB) inhibitor]. For lysed samples, plates were rinsed once in room temperature buffer containing 20 mM Hepes pH 7.4 and 150 mM NaCl and scraped in 200 µls of lysis buffer containing 20 mM Hepes, pH 7.4, 150 mM NaCl, 1% Triton X-100, 1 mM PMSF and 5 mM iodoacetamide (DUB inhibitor). Cells were incubated on ice for 5 min then spun at 21,000 ***g*** for 5 min. Supernatant was taken as the lysate.

12.5 µg of lysate was taken as a loading control. 500 µg of lysate was distributed to each condition and diluted with buffer containing 20 mM Hepes, pH 7.4, 150 mM NaCl, 1 mM PMSF and 5 mM iodoacetamide to 2 mg/ml. 20 µg of GST–GFP-Trap was added to the lysate along with 45 µl of blank GST beads. For blank-bead samples, no GST–GFP-Trap was added. The samples were rotated at room temperature for 1 h. Beads were then washed 3× in 1 ml buffer containing 20 mM Hepes, pH 7.4, 300 mM NaCl, and 0.1% Triton X-100. Protein was eluted from beads in 35 µl of 1× SDS sample loading dye, separated by SDS-PAGE and analyzed by Coomassie staining and western blotting. 20 µl of elution and lysate controls were used for western blotting to check for the presence of ARHGAP29. 10 µl of elution was used for Coomassie staining to confirm the presence of GST–GFP-Trap.

### Proximity ligation assay
MDA-MB-231 cells stably expressing GFP–Rap2a-WT or GFP–Rap2a-K3R were plated onto collagen-coated coverslips and left to adhere overnight. Cells were fixed in 10% paraformaldehyde for 10 min, quenched for 5 min (375 mg glycine in 50 ml PBS), then permeabilized with 0.1% Triton X-100 for 5 min. Following permeabilization, the Duolink proximity ligation assay (PLA) kit was used as described in the manufacturers protocol (Sigma Aldrich). Mouse anti-ubiquitin and rabbit anti-GFP primary antibodies were used together (or individually for controls). Experimental and control (GFP or ubiquitin only antibodies) cells were selected through visualization of GFP–Rap2a. Cells exhibiting front-to-back polarity with standard GFP–Rap2a fluorescence were selected for imaging. Three replicates were performed, each consisting of at least 20 cells for experimental conditions (both antibodies) or less for control conditions, which contained minimal PLA puncta.

### Incubation at 4°C and CK666 treatment
#### Incubation at 4°C
MDA-MB-231 cells stably expressing GFP–Rap2a-WT or GFP–Rap2a-K3R were seeded on collagen-coated coverslips. 24 h later, 231 medium was buffered with 20 mM Hepes, pH 7.4, and cells were divided into three groups: control, 4°C incubation and recovery. Control cells were placed in a standard incubator for 60 min before being fixed with 4% paraformaldehyde. The 4°C incubation cells were placed in a 4°C cold room for 60 min before fixing with 4% paraformaldehyde. Finally, recovery cells were placed in a 4°C cold room for 60 min, then placed in a standard incubator for 45 min before fixing with 4% paraformaldehyde.

#### CK666 treatment
MDA-MB-231 cells stably expressing GFP–Rap2a-WT or GFP–Rap2a-K3R were seeded on collagen-coated coverslips. After 24 h, cells were treated with 200 µM CK666 or DMSO (equal volume to CK666) as a control for 1 h. Cells were then fixed with 4% paraformaldehyde and processed for immunofluorescence analysis. For CK666 washout experiments, after 1 h of CK666 treatment, cells were carefully washed 3× with warmed PBS and placed with fresh medium in the incubator for 20 min to recover. Cells were then fixed with 4% paraformaldehyde.

For both experiments, cells with observable polarity were randomly selected for imaging through visualization of the actin cytoskeleton. Three biological replicates were performed, with each replicate consisting of at least 20 randomly chosen cells (technical replicates).

### Active Rap2 pulldown assays
GST–RalGDS was purified as previously described (Duncan et al., 2022). MDA-MB-231 cells stably expressing GFP–Rap2a were grown in 10 cm plates to 90% confluency, then harvested and pelleted (frozen if necessary). Pellets were lysed in buffer containing 20 mM Hepes, pH 7.4, 150 mM NaCl, 1% Triton X-100, 1 mM PMSF and 5 mM iodoacetamide (DUB inhibitor) on ice for 30 min. Lysate was clarified in chilled microcentrifuge at 21,000 ***g***. Lysate was brought to equal concentration and volume in buffer containing 20 mM Hepes pH 7.4 and 150 mM NaCl. 20 µg of either GST (control) or GST–RalGDS was added to the lysate. Lysate controls (equal µg across conditions) were also taken for later use. Tubes with the GST proteins and lysate mixture were rotated for 60 min at room temperature. 45 µl of glutathione beads (50% in PBS) were added to the tubes, and rotation was continued for 30 more minutes. Beads were then washed 5× in 1 ml buffer containing 20 mM Hepes, pH 7.4, 300 mM NaCl and 0.1% Triton X-100. Protein was eluted from beads in 35 µl of 1× SDS sample loading dye, separated by SDS-PAGE and analyzed by Coomassie staining and western blotting. 20 µl of elution and lysate controls were used for western blotting. 10 µl of elution was used for Coomassie staining to confirm the presence of GST or GST–RalGDS.

Western blots were analyzed using the gel analyzer tool in Fiji (https://fiji.sc/). Area from the pulldown was normalized to area of the lysate control to achieve a relative density measurement. Coomassie staining was not used for quantitative analysis. Data was normalized to GFP–Rap2a construct expression in the lysate control for each condition.

### CK666 treatment for active Rap2 pulldown assays
The standard Rap activity pulldown assay was altered to minimize potential GAP activity. Cells expressing GFP–Rap2a were grown in 10 cm plates. Near confluent monolayers were treated with 200 µM CK666 or an equivalent volume of DMSO for 1 h. During this time, GST–RalGDS was prebound to 45 µl of 50% GST bead slurry in 100 µl of buffer containing 20 mM Hepes pH 7.4 and 150 mM NaCl. Beads and protein were rotated at room temperature for 45 min, after which the beads were pelleted and the supernatant was removed. Prebound beads were kept on ice.

After CK666 or DMSO treatment, cell monolayers were washed with buffer containing 20 mM Hepes pH 7.4 and 150 mM NaCl. Cells were then scraped into tubes with ∼150 µl of lysis buffer containing 20 mM Hepes, pH 7.4, 150 mM NaCl, 1% Triton X-100, 1 mM PMSF and 5 mM iodoacetamide (DUB inhibitor), Tubes were placed on ice for 5 min. Lysate was clarified in chilled microcentrifuge at 21,000 ***g*** for 5 min. 400 µg of lysate was added to prebound GST–RalGDS beads and brought to an equal volume of 200 µl with buffer containing 20 mM Hepes, pH 7.4, 150 mM NaCl, 1 mM PMSF and 5 mM iodoacetamide. Tubes were rotated at room temperature for 40 min.

After rotating, beads were then washed 3× in 1 ml buffer containing 20 mM Hepes, pH 7.4, 300 mM NaCl and 0.1% Triton X-100. Protein was eluted from beads in 35 µl of 1× SDS sample loading dye, separated by SDS-PAGE and analyzed by Coomassie staining and western blotting. 20 µl of elution and lysate controls were used for western blotting. 10 µl of elution was used for Coomassie staining to confirm the presence of GST or GST–RalGDS. Data was normalized to GFP–Rap2a construct expression in the lysate control for each condition.

### cAMP treatment for immunofluorescence analysis
cAMP treatment of cells was used to test how activation and ubiquitylation of Rap2 interconnect. Cells stably expressing GFP–Rap2a-WT or GFP–Rap2a-K3R were seeded on collagen-coated coverslips. After 24 h, cells were treated with either 500 µM 8-bromo-cAMP or water (equal volume as 8-bromo-cAMP) as a control for 1 h. Cells were then taken for immunofluorescence. Polarized cells were randomly selected for imaging through visualization of the actin cytoskeleton. Three replicates were performed, with each replicate including at least 20 cells.

### Image acquisition
All fixed-cell imaging was performed on a widefield inverted Zeiss Axiovert 200M microscope using a 63× oil objective, QE charge-coupled device camera (Sensicam) and Slidebook v. 6.0 software (Intelligent Imaging Innovations). Images were taken as *z*-stacks with 0.5 µm step intervals. Where indicated, images were deconvolved (Nearest Neighbors) using the Intelligent Imaging Innovations software. SACED and GFP–paxillin

time-lapse images were acquired on the same Zeiss widefield microscope with a 63× oil objective and a temperature-controlled stage. Further image processing was performed in Fiji software.

All random migration assays were performed using OLYMPUS IX83 inverted confocal microscope, with 4× air objective, equipped with a HAMAMATSU camera controller.

Time-lapse imaging of GFP–Rap2a and RhoA biosensor dynamics were collected on a Nikon Ti2-E inverted microscope equipped with 1.45 NA 100× CFI Plan Apo objective; Nikon motorized stage; Prior NanoScan SP 600 µm nano-positioning piezo sample scanner; CSU-W1 T1 Super-Resolution spinning disk confocal; SoRa disk with 1×/2.8×/4× mag changers; 488 nm, 560 nm and 647 nm laser; and a prime 95B back illuminated sCMOS camera.

## Image analysis

Random single-cell migration analysis (Figs 1A–C and 6G,H) was performed using the Manual Tracking Excellence-Pro software (Gradientech). Cells were tracked by their geographic center. Only cells that remained in focus throughout the whole time-lapse series and did not divide were tracked. Data were generated, and parameters including speed, length and distance were acquired.

Cell morphology analysis (Figs 1E,F and 6B,C) was performed using Fiji. Images were maximum-intensity projected to include all stacks in which the actin cytoskeleton was in focus. The actin cytoskeleton from the resulting projection was traced using the polygon selection tool to create a region of interest (ROI) of the cell perimeter. The area, fitted ellipse and perimeter of the ROI was then measured. Area was defined as the area ($µm^2$), circularity as $\frac{4\pi \times area}{perimeter^2}$, and aspect ratio as the $\frac{major\ axis}{minor\ axis}$ of the fitted ellipse of the ROI.

Cortactin enrichment analysis (Fig. 1H–J) was performed using Fiji. Images were maximum-intensity projected to include all stacks in which the actin cytoskeleton was in focus. Cells with cortactin enrichment were defined as those with visible enrichment of cortactin fluorescence on a segment of the cell perimeter. Cortactin segments were measured in size by defining the percentage of the cell perimeter that they occupied. The actin cytoskeleton of the cell was traced using the polygon tool to create a ROI. The perimeter of the ROI was calculated. The segmented line tool was used to measure the length of the cortactin-enriched segment. The segments percent of the perimeter was defined as $\frac{segment\ length}{cell\ perimeter}$. Cortactin enrichment was measured by drawing ROIs around the cortactin-enriched region, a portion of the actin cortex (as defined by the actin channel) and the background fluorescence, and mean grey value of the cortactin channel was measured and used for calculations. Enrichment was defined as $\frac{cortactin\ enriched - background}{actin\ cortex - background}$. Some cells exhibited multiple cortactin-enriched segments. The percentage of the perimeter and cortactin enrichment was calculated separately for each segment. Each measurement was used when calculating the average of a replicate.

SACED analysis (Fig. 2B,C; Fig. S1B–D) was performed by creating a kymograph in Fiji. Briefly, two lines were drawn through the ruffling section of the membrane at perpendicular angles to the membrane ruffles. The exact location of the line within the ruffling section was decided randomly. Two kymographs were created using the reslice tool; one for each line. For each kymograph, measurements were taken as previously described (Hinz et al., 1999) and as defined in Fig. S1A. The measurements from each event (ruffle retraction, membrane extension/retraction) were averaged together to produce one measurement for each kymograph. The average from the kymographs were then averaged together to create an average of each measurement per cell.

GFP–Rap2a lamellipodium enrichment (Fig. S1E) was measured in Fiji. The mean grey value of GFP–Rap2a was measured both at areas of actin ruffling and at the actin cortex. Actin ruffling and the actin cortex was determined using the actin channel. GFP–Rap2a lamellipodium enrichment was defined as $\frac{actin\ ruffling\ mean\ grey\ value}{cortex\ mean\ grey\ value}$. If a cell had multiple areas of actin ruffling, enrichment was calculated for each region and the mean enrichment was reported for the cell.

GFP–Rap2a dynamics (Fig. 2F,G; Fig. S2C,D) were measured by creating kymographs in Fiji. Briefly, two lines were drawn through the ruffling section of the membrane at perpendicular angles to the membrane ruffles. The exact location of the line within the ruffling section

was decided randomly. Two kymographs were created using the reslice tool; one for each line. Enrichment from the start to end of the ruffle (Fig. 2F,G) was measured from three ruffles per kymograph. The ruffles selected were the first complete ruffle visible, the last complete ruffle visible and the ruffle most in the center of the kymograph. For each selected ruffle, a ROI was drawn around approximately the first third and last third of the ruffle. Mean grey value was measured from each ROI, and the background mean grey value was subtracted from each measurement. The average mean grey value was found for each cell. Enrichment was defined as the $\frac{average\ ruffle\ start\ mean\ grey\ value}{average\ ruffle\ end\ mean\ grey\ value}$. Ruffle retraction dynamics (Fig. S2C,D) were measured from the GFP–Rap2a using the same measurements as the SACED dynamics.

RhoA biosensor ruffle enrichment and longevity (Fig. 3D,E) were measured by drawing a line through ruffles with the RhoA biosensor. Lines were drawn through three ruffles per cell – one at the start, middle and end of each video. Ruffles were only selected if they were distinct from adjacent ruffles, and lines were drawn through ruffles on the frame when the ruffle was at its brightest fluorescence by eye. Line scans of fluorescence intensity were taken and enrichment was defined as $\frac{peak\ flourescence\ intensity\ of\ ruffle}{first\ valley\ fluorescence\ intensity\ of\ ruffle}$. Ruffle longevity was measured for the ruffles of which enrichment was measured. However, ruffles were excluded if they started or ended outside the video time. The number of frames in which the RhoA sensor was present and enriched in the ruffle were counted and multiplied by five to calculate the time in seconds.

ARHGAP29 line scans (Fig. 4C,D,F) were performed by making a z-projection from the images of a z-stack where the lamellipodium was in focus. A line was drawn across an area of interest (in the direction shown by the arrow in the corresponding image), and the plot profile tool in Fiji was used to extract the fluorescence intensity along the line. The maximum intensity for each channel was used to normalize the fluorescence intensity. Multiple line scans from one cell (Fig. 4C,D) used the same maximum intensity per channel.

To analyze focal adhesion dynamics (Fig. 5A–D), the TrackMate plugin in Fiji was used to manually track GFP-positive puncta (FAs). Puncta were tracked across each frame until they disappeared. Data for each puncta were generated by TrackMate. Kymographs were generated by drawing a line across a leading edge-localized GFP puncta and using the reslice tool.

Focal adhesion count, size and periphery localization (Figs 5F–H and 6E,F) was measured in Fiji. Up to three stacks where the phospho-paxillin signals (focal adhesions) were in focus were maximum-intensity projected. Images were cropped to only include one cell at a time, and further analysis was performed without reference to the sample identity. The phospho-paxillin channel was thresholded to apply a mask and create puncta for all focal adhesions in a cell. The analyze particle tool was then used to measure the size and number of puncta. Data were filtered to only include puncta larger than $0.04\ µm^2$ and less than $2\ µm^2$. Filtered puncta size was averaged per cell. Puncta number and average size per cell was averaged across all cells to create the mean replicate value. For measuring the localization of puncta, a ROI was drawn around the outside of the cell using the actin channel. The enlarge tool was used to then shrink the ROI by 2.5 µm to make an intracellular ROI. Percent of puncta area in the periphery was calculated as $\frac{whole\ cell\ puncta\ area - intracellular\ puncta\ area}{whole\ cell\ puncta\ area} \times 100$.

MyoIIb analysis (Fig. 5J) was performed in Fiji. Z-stacks were maximum-intensity projected to include the two or three frames where stress fibers were in focus. An intracellular ROI was drawn using the actin channel (excluding the actin cortex). The actin channel was modified by removing the background using a 5.0 rolling ball radius, and all fluorescence outside the intracellular ROI was cleared. A threshold was made on the modified actin channel to create a mask of the acto-myosin stress fibers. The mask was translated −3 pixels on the y-axis to account for dichromatic shift. The mean grey value of the intracellular ROI and stress fiber mask was measured on the MyoIIb channel. MyoIIb enrichment on the stress fibers was defined as $\frac{stress\ fiber\ mask\ MyoIIb\ mean\ grey\ value}{intracellular\ ROI\ MyoIIb\ mean\ grey\ value}$.

PLA analysis (Fig. 7C–E) was performed using Fiji. All z-stack images were maximum-intensity projected to include all PLA puncta except puncta on the cell ventral surface. The images were then cropped to only show one cell at a time. The polygon tool was used to trace the outside of the GFP–Rap2a signal, creating a whole-cell ROI. A threshold was applied to the PLA

channel to make masks of PLA puncta. Thresholding was done to minimize any residual signal from ventral and ventral surface PLA puncta. The analyze particle tool was used to count the number of puncta greater than 0.04 μm$^2$ in the whole-cell ROI. Whole-cell area was calculated from the whole-cell ROI. To analyze membrane-localized PLA puncta, this process was repeated using a whole-cell and an intracellular (defined by GFP–Rap2a channel) ROI. PLA puncta were counted, and the percent of membrane-localized PLA puncta was defined as $\frac{\text{whole cell puncta} - \text{intracellular puncta}}{\text{whole cell puncta}} \times 100$. PLA puncta enrichment in membrane ruffles was calculated by using a whole-cell ROI and multiple ROIs per cell defining membrane ruffling (defined by GFP–Rap2a channel). Whole-cell puncta density was defined as $\frac{\text{number of PLA puncta}}{\text{whole} - \text{cell ROI area}}$. Membrane ruffling puncta density was defined as $\frac{\sum (\text{PLA puncta in membrane ruffles})}{\sum (\text{membrane ruffles ROI area})}$.

Intracellular/whole-cell fluorescence ratio (Figs 8D,F and 9B,J; Fig. S4B) was calculated in Fiji using images that were maximum-intensity projected on the stacks where GFP–Rap2a signal was in focus. Per cell, the actin channel was used to define the cell perimeter. An ROI was traced on the outside of the actin channel and defined as the whole cell, while another ROI was traced on the inside of the actin channel and defined as the intracellular ROI. The ROIs were then applied to the GFP–Rap2a channel, and the integrated density was measured. ROIs were also used to measure the integrated density of the background fluorescence. Data was expressed as a ratio of integrated density, which was defined as $\frac{\text{intracellular} - \text{background}}{\text{whole} - \text{background}}$, where a value of 1 would indicate all the signal was intracellular.

## Statistical analysis

Graphs are displayed as biological replicates (opaque dots) overlayed on technical replicates (translucent dots). Error bars represent the standard deviation of the biological replicates. Replicates are displayed by color. Statistical analysis was performed on biological replicates (when possible), which were calculated from the mean of the technical replicates. Few analyses (Fig. 5H,J) were performed on technical replicates. For all analyses comparing three or more conditions, a one-way ANOVA was used with comparisons run for each column against the mean of each other column. All significant *P*-values are displayed on graphs. For all analyses comparing two conditions, a Student's *t*-test was used, with all significant *P*-values displayed. A paired *t*-test was used for Fig. 2F and unpaired *t*-tests were used for all other *t*-tests.

## Acknowledgements

Thanks to the Garcia-Matta lab for providing the dTom-2xrGBD RhoA biosensor. Thank you to Jeff Moore and Jeremy Brown for use of the spinning disk confocal microscope. Thank you to Paige Neumann for grammatical correction of this manuscript.

## Competing interests

The authors declare no competing or financial interests.

## Author contributions

Conceptualization: A.N., R.S., R.P.; Data curation: A.N., E.M.; Formal analysis: A.N., R.S.; Funding acquisition: V.A.S., R.P., A.N.; Investigation: A.N., R.S., E.M.; Methodology: A.N., R.P.; Visualization: A.N.; Writing – original draft: A.N., R.S., R.P.; Writing – review & editing: A.N., R.S., V.M., V.A.S., I.S., R.P.

## Funding

This work was funded by the National Institutes of Health grant R01 GM122768 (to R.P.), grant S-MIP-22-60 from Research Council of Lithuania (to R.P.), and National Institutes of Health grant T32 GM136444 (to A.N.). Open Access funding provided by University of Colorado. Deposited in PMC for immediate release.

## Data and resource availability

All relevant data and details of resources can be found within the article and its supplementary information. Any material (plasmids, cell lines) generated in this paper can be obtained by emailing the corresponding author.

## First Person

This article has an associated First Person interview with the first author of the paper.

## Peer review history

The peer review history is available online at https://journals.biologists.com/jcs/lookup/doi/10.1242/jcs.264375.reviewer-comments.pdf

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
