## [Peer Review File · Journal of Cell Science]

Ubiquitylation-dependent Rap2 activation regulates lamellipodia dynamics during cell migration

Andrew Neumann, Revathi Sampath, Emily Mayerhofer, Valeryia Mikalayeva, Vytenis Arvydas Skeberdis, Ieva Sarapinienė and Rytis Prekeris
DOI: 10.1242/jcs.264375

Editor: Guillaume Jacquemet

Review timeline

Original submission:	12 August 2025
Editorial decision:	2 September 2025
First revision received:	29 September 2025
Editorial decision:	13 October 2025
Second revision received:	28 October 2025
Accepted:	30 October 2025

Original submission

First decision letter

MS ID#: jcs.264375

MS TITLE: Mono-Ubiquitylation-Dependent Rap2 Activation Regulates Lamellipodia Dynamics During Cell Migration

AUTHORS: Andrew Neumann; Revathi Sampath; Emily Mayerhofer; Valeryia Mikalayeva; Vytenis Arvydas Skeberdis; Ieva Sarapinienė; Rytis Prekeris

ARTICLE TYPE: Research Article

Dear Dr Prekeris,

We have now reached a decision on the above manuscript.

To see the reviewers' reports and a copy of this decision letter, please go to: View Reviewer Comments

Reviewer 1

Advance summary and potential significance to field

In their manuscript "Mono-Ubiquitylation-Dependent Rap2 Activation Regulates Lamellipodia Dynamics During Cell Migration", Neumann et al., have suggested a very interesting mechanism by which Rap2 influences cell migration in MDA cells via modulating RhoA activity at the leading edge during the retraction of lamellipodia. This mechanism is supported by extensive and rigorous data using a range of biochemical and microscopy approaches, and has exciting applicability in the wider field beyond Rap2 when considering the possibilities and implications of mono-ubiquitylation of other widely studied small GTPases such as Rho, Rab, Ras and Rac. I commend the authors for their honest highlighting of what they've shown in this manuscript, and what mechanistically still remains

to be determined. I believe that this manuscript is potentially suitable for publication at the journal of cell science if the following major comments are addressed and quantifications added:

Comments for the author

Major comments [Please request additional experiments only if they are essential for supporting the conclusions; authors should be encouraged to highlight any claims that are preliminary or speculative, or to discuss any pitfalls or alternative interpretations in a 'Limitations' section]

1. One important role of RhoA at the leading edge which has not been mentioned at all in the manuscript is in promoting protrusions via formins such as mDia1 (generally filopodial protrusions but also in lamellipodia as in <https://pmc.ncbi.nlm.nih.gov/articles/PMC3116607/> , this is T cells but there is evidence in multiple cell types) rather than just contracting protrusions as mentioned here. Can the authors comment on if Rap2 may have a role in regulating the RhoA-formins pathway leading to protrusions, and if they don't think this has any role can they say why? While ideally the authors would image formins fixed or live in their Rap2 KO cells or use a formin inhibitor to check whether all their other findings hold true in formin perturbed conditions, it may be beyond the scope of this work but I would still like to see some content added to the discussion on this additional role of RhoA.

2. As mentioned by the authors, the canonical role of RhoA is in contractility/retraction of the rear/uropod/trailing edge. Have the authors looked at whether their Rap2 mechanism also plays a role in regulating RhoA contractility here, and if it does not, why is this Rap2-RhoA pathway restricted to the leading edge and the retraction of the lamellipodium phase specifically? Do the authors have live images as in Figure 2D where the whole cell including the rear is visible? Other images in the manuscript showing whole cell localisation of Rap2 (e.g. 4C) indicate there is Rap2 (and ARHGAP29) at both edges of the cell, unless this is an unpolarised cell with lamellipodia on both sides. I think it would help to comment in the discussion on any potential role for Rap2-ARHGAP29 regulating RhoA at the trailing edge membrane as well (note this is a different role for RhoA than in stress fiber formation across the middle of the cell).

3. Could the authors elaborate more on the use of hypotonic shock in the pull downs for figure 4 as I could not see any mention of this hypotonic shock in the methods. Hypo and hyper osmotic shock can have significant effects on RhoA activity (likely by altering membrane tension <https://pmc.ncbi.nlm.nih.gov/articles/PMC6863396/>) and so I am not sure if the Rap2-ARHGAP29 interaction may only be occurring in this particular perturbed hypotonic case and not in isotonic conditions, however this may be easier to interpret if the hypertonic shock method is added to the methods. Did the authors try the same experiments in isotonic conditions and if not, why not? If technically possible, I would suggest to repeat the experiment in figure 4 in isotonic conditions.

4. Some additional quantification and statistical analysis is required throughout the manuscript. In particular, Rap2 localisation to the lamellipodium is a crucial finding of the work and is frequently mentioned in the text without accompanying quantification and formal analysis:

- Figure 2D, 3A: Rap2 localisation should be quantified (e.g. is it enriched at the leading edge?)
- Figure 4B, C: ARHGAP29 and Rap2 localisation should be quantified (e.g. using line profiles from front to rear)
- Figure 3A, 4C: colocalisation analysis between RhoA/Rap2 and ARHGAP29/Rap2 would help
- Figure 5E: quantification of peripheral and central p-pax adhesions should be performed and compared between control and Rap2 KO cells and tested for significance
- Figure 5H, I: The average MyoIIB across multiple cells should be measured and compared for control and Rap2 KO cells and tested for significance
- Figure 7A: A key finding of the manuscript is that increased PLA puncta are found in the lamellipodium, so this needs to be carefully analysed and quantified to show lamellipodial puncta in WT vs K3R cells (either manually identifying the lamellipodia or using a specific marker such as cortactin as in figure 1)
- Figure 9A: Total and membrane Rap2 should be quantified and compared for each condition
- Figure 9I: as above (7A), lamellipodial specific Rap2 should be quantified and compared in each condition in addition to the intracellular/whole cell quantification

5. Linked to the previous point, the authors must state their number of cells used/quantified and number of biological/experimental replicates performed for all their experiments (particularly the imaging) as presently it is impossible to tell if some observations were repeated

Minor comments

'Lamellipodium' is the singular form however 'lamellipodia' is frequently used throughout the manuscript in its place when talking about the single leading edge protrusion

Reviewer 2

Advance summary and potential significance to field

The main research question addressed in this manuscript is how Rap2 functions and how its ubiquitination determines its localisation to lamellipodia. This is a continuation of a study published by the same research group, where it was shown that ubiquitination by Rab40b/Cul5 regulates Rap2 localization and activity during cell migration (Duncan et al 2022). In this study, it is shown that Rap2 deficiency slows migration by affecting leading edge formation. This is suggested to be caused by lamellipodia structures not being properly formed, which affects the polarity of migrating cells. It is suggested that Rap2 functions as a RhoA inhibitor in retracting lamellipodia ruffles and that loss of Rap2-mediated inhibition of RhoA leads to defects in focal adhesions disassembly. To study the effect of ubiquitination on Rap2 function and localisation, the authors expressed a previously published K3R mutant, in which three ubiquitination sites of Rap2 were mutated. They found that the Rap2K3R mutant failed to rescue the loss of front-to-back polarity and ability to migrate induced by loss of Rap2. Furthermore, when forcing the Rap2 K3R mutant to the membrane, it was not able to further increase its constitutive activity, while WT was able to. The data did not provide information about how ubiquitination affects Rap2 localisation, but the authors conclude that Rap2 ubiquitination regulates Rap2 activity, which in turn mediates its localisation to the plasma membrane. As Rap2 seems to be a significant regulator of lamellipodia and cell migration, it would be of importance to understand the molecular mechanisms regulating Rap2 activity. Indeed, in this study, the authors aim to provide mechanistic data to a previous publication, where they show that ubiquitination determines its localisation to lamellipodia. While new information of activation of Rap2 is provided, the study fails to describe how ubiquitination affects Rap2 localisation. The manuscript is well written, and most of the data presented is of good quality, but some of the experiments lack proper controls.

Comments for the author

Major comments

* The major focus of this study is to examine how loss of mono-ubiquitination affects Rap2 function and localisation. However, it is only assumed, not experimentally shown that Rap2 is mono-ubiquitinated. In addition, this study use a K3R mutant to study the effect of mono-ubiquitination of Rap2, although it is not shown (here or previously) that this mutation prevents mono-ubiquitination of Rap2. Furthermore, PLA using a pan-ubiquitin antibody does not show mono-ubiquitination, it only show association with ubiquitin.

* The expression of Rap2 K3R seems very low in Fig 6A compared to WT Rap2. Likewise, does the overexpressed K3R mutant seem to be expressed at lower level compared to WT in other experiments (Fig 8G and Fig S3). The authors need to accomplish similar expression levels of WT and K3R mutants to confirm that the lost effects depend on the mutation and not the expression level.

* Only Fig 6 shows data where WT and K3R Rap2 are used to rescue a KO phenotype, the other are overexpression studies. Protein expression levels can affect both activity and localisation of proteins. Hence, changes in localisation and activity needs to be studied also on endogenous proteins or at least in a rescue system where it is shown to be expressed at endogenous level. In addition, the expression levels of WT and K3R must be comparable.

Minor comments

- * Although the change in Paxilling punctae is convincingly shown in the left panels in Fig 6D, the right panels showing GFP-Rap2+Merge are not.
- * The ability to migrate of the K3R mutation should be compared to KO, not only to WT rescue (Fig 6G).
- * The authors claim that PLA puncta in cells expressing GFP-Rap2a-WT can be observed on the edge of the cell and on the ventral surface of the cell membrane with only a few punctae found in the cytoplasm. However, in Fig 7A and B, the punctae look evenly distributed throughout the cell. In addition, in this experiment, WT and K3R Rap2 seem similarly distributed in the cell, differing from other shown images.
- * Most figures are very bad resolution. Some (Fig 8) so bad that the texts cannot be distinguished.

First revision

Author response to reviewers' comments

We would like to thank the Reviewers for very constructive comments and suggestions. In this revised manuscript we incorporated most of them and we believe that this enhanced the manuscript. The point-by-point rebuttal is listed below. All text changes in manuscript are marked in yellow.

Reviewer 1

Major comments:

1. One important role of RhoA at the leading edge which has not been mentioned at all in the manuscript is in promoting protrusions via formins such as mDia1 (generally filopodial protrusions but also in lamellipodia as in <https://pmc.ncbi.nlm.nih.gov/articles/PMC3116607/>, this is T cells but there is evidence in multiple cell types) rather than just contracting protrusions as mentioned here. Can the authors comment on if Rap2 may have a role in regulating the RhoA-formins pathway leading to protrusions, and if they don't think this has any role can they say why? While ideally the authors would image formins fixed or live in their Rap2 KO cells or use a formin inhibitor to check whether all their other findings hold true in formin perturbed conditions, it may be beyond the scope of this work but I would still like to see some content added to the discussion on this additional role of RhoA.

We agree that there is a possibility for Rap2 and ARHGAP29 to be regulating the RhoA-formin pathway, either directly or indirectly. However, we believe that thoroughly addressing this possibility lies outside the scope of our question for this paper. As such, we have added a paragraph to the discussion highlighting how formins function in the lamellipodium and, as a result, how Rap2 loss might influence these functions either directly or as a result of large scale RhoA activity disruption.

2. As mentioned by the authors, the canonical role of RhoA is in contractility/retraction of the rear/uropod/trailing edge. Have the authors looked at whether their Rap2 mechanism also plays a role in regulating RhoA contractility here, and if it does not, why is this Rap2-RhoA pathway restricted to the leading edge and the retraction of the lamellipodium phase specifically? Do the authors have live images as in Figure 2D where the whole cell including the rear is visible? Other images in the manuscript showing whole cell localization of Rap2 (e.g. 4C) indicate there is Rap2 (and ARHGAP29) at both edges of the cell, unless this is an unpolarized cell with lamellipodia on

both sides. I think it would help to comment in the discussion on any potential role for Rap2-ARHGAP29 regulating RhoA at the trailing edge membrane as well (note this is a different role for RhoA than in stress fiber formation across the middle of the cell).

While RhoA regulation at the lagging edge could be regulated by Rap2-ARHGAP29, we have chosen to focus on the leading edge as ARHGAP29 fluorescent intensity is greater in the lamellipodium than the lagging edge. We have added line scans (Fig. 4C-D) to help clarify this point. Furthermore, while Rap2 can be seen to be localized at the membrane of lagging edge (Fig. 3A), the image in Fig. 4E is that of a multipolar cell with multiple lamellipodia. This issue has been clarified in the text. Thus, we have described these observations and explained why we chose to focus on leading edge Rap2-ARHGAP29 in the text.

3. Could the authors elaborate more on the use of hypotonic shock in the pull downs for figure 4 as I could not see any mention of this hypotonic shock in the methods. Hypo and hyper osmotic shock can have significant effects on RhoA activity (likely by altering membrane tension <https://pmc.ncbi.nlm.nih.gov/articles/PMC6863396/>) and so I am not sure if the Rap2-ARHGAP29 interaction may only be occurring in this particular perturbed hypotonic case and not in isotonic conditions, however this may be easier to interpret if the hypertonic shock method is added to the methods. Did the authors try the same experiments in isotonic conditions and if not, why not? If technically possible, I would suggest to repeat the experiment in figure 4 in isotonic conditions.

We have included the methods for hypotonic shock in the methods titled as “GFP-Rap2a and ARHGAP29 binding assay”. We have shown that this method is technically necessary as we believe that membrane binding is needed (thus detergents cannot be used to lyse the cell) for ARHGAP29 and GFP-Rap2a have no interaction. Repeating the assay with hypotonic shock induced bursting of cells allows for membrane fragments to be present in the assay. ARHGAP29 has multiple predicted membrane binding domains (f-BAR and DAG binding). We suspect that these domains interacting with the membrane is necessary for proper ARHGAP29 structure and allow for Rap2 binding. These speculations are now included in the text.

4. Some additional quantification and statistical analysis is required throughout the manuscript. In particular, Rap2 localization to the lamellipodium is a crucial finding of the work and is frequently mentioned in the text without accompanying quantification and formal analysis:

- Figure 2D, 3A: Rap2 localization should be quantified (e.g. is it enriched at the leading edge?). *We have quantified GFP-Rap2a lamellipodia enrichment as compared to the cortex using our fixed cell imaging. Data has been added to Figure S1E.*

- Figure 4B, C: ARHGAP29 and Rap2 localization should be quantified (e.g. using line profiles from front to rear). *We have used a line-scan to show the ARHGAP29 and Rap2 localization in Figure 4F.*

- Figure 3A, 4C: colocalization analysis between RhoA/Rap2 and ARHGAP29/Rap2 would help. *ARHGAP29 and Rap2 localization has been quantified using a line scan in Figure 4F. Colocalization between the RhoA biosensor and Rap2 has been shown over time by merging a kymograph in Figure 3B. A more direct quantification of colocalization (Pearson’s correlation coefficient, Mander’s coefficient) is not possible as the cytoplasmic staining of ARHGAP29 and dim fluorescence of the RhoA biosensor prevents accurate thresholding of those channels.*

- Figure 5E: quantification of peripheral and central p-pax adhesions should be performed and compared between control and Rap2 KO cells and tested for significance. *We quantified peripheral p-pax puncta area (Fig. 5H) and found data, that there is more p-pax puncta area in the periphery of Rap2-KO cells, that was contrary to our original claim. As such, we have altered the text and our interpretation of our results now reads to reflect this quantification.*

- Figure 5H, I: The average MyoIIb across multiple cells should be measured and compared for control and Rap2 KO cells and tested for significance. *We have quantified the fluorescent enrichment of MyoIIb on stress fibers as compared to the whole cell. The quantification and new images are now shown in Figure 5J.*

- Figure 7A: A key finding of the manuscript is that increased PLA puncta are found in the lamellipodium, so this needs to be carefully analyzed and quantified to show lamellipodial puncta in WT vs K3R cells (either manually identifying the lamellipodia or using a specific marker such as cortactin as in figure 1). *As suggested, we have added two quantifications to address this point. The localization of PLA puncta in GFP-Rap2a-WT vs GFP-Rap2a-K3R expressing cells to the plasma membrane has been quantified (Fig. 7D). Further, we compared the density of PLA puncta in lamellipodia ruffles to the density PLA signal in entire cell (Fig. 7E).*

- Figure 9A: Total and membrane Rap2 should be quantified and compared for each condition. *Figure 9B has been added showing quantification of Figure 9A.*

- Figure 9I: as above (7A), lamellipodial specific Rap2 should be quantified and compared in each condition in addition to the intracellular/whole cell quantification. *Figure 9J has been added that shows quantification of Figure 9I.*

5. Linked to the previous point, the authors must state their number of cells used/quantified and number of biological/experimental replicates performed for all their experiments (particularly the imaging) as presently it is impossible to tell if some observations were repeated. *We have added in the figure legends the number of cells and replicates used in our experiments.*

Minor comments:

1) 'Lamellipodium' is the singular form however 'lamellipodia' is frequently used throughout the manuscript in it's place when talking about the single leading edge protrusion.

Lamellipodia has been changed to lamellipodium throughout the text where appropriate.

Reviewer 2

Major comments:

1) The major focus of this study is to examine how loss of mono-ubiquitination affects Rap2 function and localization. However, it is only assumed, not experimentally shown that Rap2 is mono-ubiquitinated. In addition, this study used a K3R mutant to study the effect of mono-ubiquitination of Rap2, although it is not shown (here or previously) that this mutation prevents mono-ubiquitination of Rap2. Furthermore, PLA using a pan-ubiquitin antibody does not show mono-ubiquitination, it only shown association with ubiquitin.

The reviewer is correct that while our past work is highly suggestive that the Rab40/CRL5 complex mono-ubiquitylates Rap2a at K117, K148, and K150, it is not definitively conclusive that mono-ubiquitylation is what occurs at those three Lysine residues. Regardless, our past data does show that the K3R mutant does stop Rab40/CRL5 ubiquitylation of Rap2. Understanding the function and the series of events of activation and localization that result from Rab40/CRL5 ubiquitylation remains the main point of this paper. As such, we have provided this clarification in the introduction and in most places replaced term "mono-ubiquitylation" with "ubiquitylation".

2) The expression of Rap2 K3R seems very low in Fig 6A compared to WT Rap2. Likewise, does the overexpressed K3R mutant seem to be expressed at lower level compared to WT in other experiments (Fig 8G and Fig S3). The authors need to accomplish similar expression levels of WT and K3R mutants to confirm that the lost effects depend on the mutation and not the expression level.

The reviewer is correct that GFP-Rap2a-WT and GFP-Rap2a-K3R do not express at equal levels. Our past work showed that when Rap2 is unable to be ubiquitylated by the Rab40/CRL5 complex, it is localized to the lysosome and is degraded rapidly, to the point that by passage 5, notable loss of Rap2 expression occurs. Accordingly, we prioritize using cells in early passages and compare early passage to later passage data to ensure that decreased expression as passaging continues does not alter our data. We have added a statement to the text addressing this issue in the "Generation of lentiviral stable cell lines" section of the methods.

3) Only Fig 6 shows data where WT and K3R Rap2 are used to rescue a KO phenotype, the other are overexpression studies. Protein expression levels can affect both activity and localization of

proteins. Hence, changes in localization and activity needs to be studied also on endogenous proteins or at least in a rescue system where it is shown to be expressed at endogenous level. In addition, the expression levels of WT and K3R must be comparable.

The reviewer is correct again that GFP-Rap2a-WT and GFP-Rap2a-K3R do not express at the same levels for the reasons listed above. We treated these cells as our other stable lines (using low passages) in order to ensure that expression is not what is driving our phenotype rescue. We also took the reviewers advice and added a western blot in Figure 6I of our rescue lines as compared to endogenous Rap2 in our control lines to show that even GFP-Rap2a-K3R lower expression is still greater than endogenous Rap2 expression and commented on this difference in expression in the text. Additionally, we updated the GFP-Rap2a-K3R rescue images in Figure 6A and 6D.

Minor comments:

1) Although the change in Paxillin punctae is convincingly shown in the left panels in Fig 6D, the right panels showing GFP-Rap2+Merge are not.

We have changed the image in Figure 6D with the image that better reflects the data.

2) The ability to migrate of the K3R mutation should be compared to KO, not only to WT rescue (Fig 6G).

We agree that this experiment would be optimal. We have done these experiments in our collaborator's laboratory in University of Health Sciences in Lithuania (since we do not have suitable microscopy set-up here at University of Colorado). Unfortunately, shipping live cells from USA to Lithuania become very difficult due to latest changes in shipping rules (both in US and EU), thus, we cannot perform suggested analysis. Instead, in new version of the manuscript we referenced the migration data from Rap2-KO cells in Figure 1B-C in the text and compared it to the Rap2-KO + GFP-Rap2a-K3R data from Figure 6G-H.

3) The authors claim that PLA puncta in cells expressing GFP-Rap2a-WT can be observed on the edge of the cell and on the ventral surface of the cell membrane with only a few punctae found in the cytoplasm. However, in Fig 7A and B, the punctae look evenly distributed throughout the cell. In addition, in this experiment, WT and K3R Rap2 seem similarly distributed in the cell, differing from other shown images.

We have added two quantifications to address this point. The localization of PLA puncta in GFP-Rap2a-WT vs GFP-Rap2a-K3R expressing cells at the plasma membrane has been quantified (Fig. 7D). Further, we compared the densities of PLA puncta in lamellipodia ruffles and whole cell (Fig. 7E). These new quantifications clearly demonstrate that PLA signal is enriched at the plasma membrane and lamellipodia. Additionally, we changed the image in Fig. 7A-B for the GFP-Rap2a-K3R expressing cell to better represent our new data.

4) Most figures are very bad resolution. Some (Fig 8) so bad that the texts cannot be distinguished.

We have reuploaded higher resolution figures.

Second decision letter

MS ID#: jcs.264375R1

MS TITLE: Ubiquitylation-Dependent Rap2 Activation Regulates Lamellipodia Dynamics During Cell Migration

AUTHORS: Andrew Neumann; Revathi Sampath; Emily Mayerhofer; Valeryia Mikalayeva; Vytenis Arvydas Skeberdis; Ieva Sarapinienė; Rytis Prekeris

ARTICLE TYPE: Research Article

Dear Dr Prekeris,

We have now reached a decision on the above manuscript.

To see the reviewers' reports and a copy of this decision letter, please go to:

As you will see, the reviewer 1 gave a favourable report, but reviewer 2 raised some critical points that will require amendments to your manuscript. I hope that you will be able to carry these out because I would like to be able to accept your paper, depending on further comments from reviewers.

When addressing reviewer 2's original comments, please pay special attention to demonstrating that the expression level of Rap2 K3R is similar to that of WT when using the cell line with low passages. I also encourage you to tone down some of the conclusions and to include a section on the study's limitations in the discussion.

Reviewer 1

Advance summary and potential significance to field

I believe that the authors have substantially and comprehensively responded to all my initial major and minor comments in my original review and as such this manuscript is suitable for publication in JCS without any further revisions needed.

Reviewer 2

Advance summary and potential significance to field

The main research question addressed in this manuscript is how Rap2 functions and how its ubiquitination determines its localisation to lamellipodia. This is a continuation of a study published by the same research group, where it was shown that ubiquitination by Rab40b/Cul5 regulates Rap2 localization and activity during cell migration (Duncan et al 2022). In this study, it is shown that Rap2 deficiency slows migration by affecting leading edge formation. This is suggested to be caused by lamellipodia structures not being properly formed, which affects the polarity of migrating cells. It is suggested that Rap2 functions as a RhoA inhibitor in retracting lamellipodia ruffles and that loss of Rap2-mediated inhibition of RhoA leads to defects in focal adhesions disassembly. To study the effect of ubiquitination on Rap2 function and localisation, the authors expressed a previously published K3R mutant, in which three ubiquitination sites of Rap2 were mutated. They found that the Rap2K3R mutant failed to rescue the loss of front-to-back polarity and ability to migrate induced by loss of Rap2. Furthermore, when forcing the Rap2 K3R mutant to the membrane, it was not able to further increase its constitutive activity, while WT was able to. The data did not provide information about how ubiquitination affects Rap2 localisation, but the authors conclude that Rap2 ubiquitination regulates Rap2 activity, which in turn mediates its localisation to the plasma membrane. As Rap2 seems to be a significant regulator of lamellipodia and cell migration, it would be of importance to understand the molecular mechanisms regulating Rap2 activity. Indeed, in this study, the authors aim to provide mechanistic data to a previous publication, where they show that ubiquitination determines its localisation to lamellipodia. While new information of activation of Rap2 is provided, the study fails to describe how ubiquitination affects Rap2 localisation. The manuscript is well written, and most of the data presented is of good quality, but some of the experiments lack proper controls.

Comments for the author

The authors have responded to the reviewer comments by giving explanations and possible causes for the observations criticised. However, they have not addressed the concerns experimentally.

They have neither appropriately addressed the experimental weaknesses in the manuscript. Furthermore, in their rebuttal, the authors claim that it previously has been shown that K3R mutation of Rap2 abolishes Rap2 ubiquitination. However, in the manuscript they only refer to Duncan et al 2022, where this was not shown (only localisation and binding to Rab40b were analysed). Similarly, many of the current results are misinterpreted. For example, as pointed out in the first revision, the expression level of Rap2 K3R is lower than WT. The authors do logically attribute this to the K3R mutant is degraded via autophagy. This may be true, but in their experiments, they express K3R and claims that it fails to rescue phenotypes of Rap2-KO lines (such as the front-to-back polarity). If K3R is not expressed in the cells, regardless of the reason, it cannot rescue an effect. Hence, phenotypes can only be attributed to the expression, not to a possible ubiquitin modification. Likewise, if the activity between WT and K3R is compared, the protein amounts must be comparable.

Second revision

Author response to reviewers' comments

Reviewer #2

1) In their rebuttal, the authors claim that it previously has been shown that K3R mutation of Rap2 abolishes Rap2 ubiquitination. However, in the manuscript they only refer to Duncan et al 2022, where this was not shown (only localization and binding to Rab40b were analyzed).

Reviewer is correct that we did not unequivocally demonstrate that K3R mutation blocks Rap2 ubiquitylation. We would like, however, to point out that there are many lines of evidence that strongly implies that (see below).

1) We have directly shown that Rap2 is ubiquitylated by Rab40/Cul5 complex (Duncan et al paper).

2) We have demonstrated that Rap2 binds to Rab40/Cul5 complex and that mutations in Rab40 that blocks its ubiquitylation activity increases binding (what is expected from enzyme-substrate interaction). Importantly, we show that K3R mutation within Rap2 (presumably blocking its ubiquitylation) also increases Rap2 binding to K3R. All these data are shown in Duncan et al paper.

3) Defects in Rap2-K3R localization fully phenocopies the effect of Rab40 knock out on WT-Rap2 localization.

4) Rab40 knock out or K3R mutation within Rap2 leads to similar decrease in Rap2 activation (this study and Duncan et al paper).

5) Rab40 knock out and Rap2 knock out leads to similar defects in cell migration that cannot be rescued with Rap2-K3R mutant (this study).

6) PLA signal that presumably detects ubiquitylated Rap2 significantly decreased in Rap2-K3R expressing cells (this study).

*While all these data strongly implies that K3R mutation blocks Rab40-dependent ubiquitylation, as suggested, we added **Study Limitations** chapter (to Discussion section) to list all the limitations of our experiments, including the questions of whether K3R blocks ubiquitylation.*

2) Similarly, many of the current results are misinterpreted. For example, as pointed out in the first revision, the expression level of Rap2 K3R is lower than WT. The authors do logically attribute this to the K3R mutant is degraded via autophagy. This may be true, but in their experiments, they express K3R and claims that it fails to rescue phenotypes of Rap2-KO lines (such as the front-to-back polarity). If K3R is not expressed in the cells, regardless of the reason, it cannot rescue an effect. Hence, phenotypes can only be attributed to the expression, not to a possible ubiquitin modification. Likewise, if the activity between WT and K3R is compared, the protein amounts must be comparable.

*We fully agree with the reviewer that if the function of WT and K3R is compared, cells must express the protein amounts that is comparable. Since cells tend to degrade K3R more efficiently, in our experiments we did go an extra way to ensure that expression levels were comparable. For example, we started experiments by using only low passage K3R-expressing cells since in later passages K3R levels tend to decrease. As passage progressed, data did not vary (see data distribution in graphs), indicating that K3R expression levels are not driving our results. Our apologies if we did not make that clear in previous version of the manuscript. In this version of the manuscript, we now show two western blots making it clear that similar expression of WT and K3R constructs occurs in early passage cells. **First blot** (Figure 6I) shows that WT and K3R are expressed in comparable levels in KO cells. Thus, lack of rescue by K3R is not due to differences in expression but due to mutations that presumably block Rap2 ubiquitylation. **Second blot** (Supplemental Figure 3C) shows WT and K3R expression levels in wild type MDA-MB-231 cells. These cells were used for all our localization and activity assays. Again, the differences in Rap2 activity and function are clearly caused by mutation but not differences in expression levels. Finally, in all our Rap2 activity pull-down assays, we always normalized the levels of active Rap2 (that is present in the pull-down fraction) to the levels of total cellular Rap2.*

Third decision letter

MS ID#: jcs.264375R2

MS Title: Ubiquitylation-Dependent Rap2 Activation Regulates Lamellipodia Dynamics During Cell Migration

Authors: Andrew Neumann; Revathi Sampath; Emily Mayerhofer; Valeryia Mikalayeva; Vytenis Arvydas Skeberdis; Ieva Sarapinienė; Rytis Prekeris

Article Type: Research Article

Dear Dr Prekeris,

I am happy to tell you that your manuscript has been accepted for publication in Journal of Cell Science, pending standard publication integrity checks.